# Triple-BERT: Do We Really Need MARL for Order Dispatch on Ride-Sharing Platforms?

**Zijian Zhao**[1], **Sen Li**[1,2] *
[1]The Hong Kong University of Science and Technology
[2]The Hong Kong University of Science and Technology (Guangzhou)

## Abstract

On-demand ride-sharing platforms, such as Uber and Lyft, face the intricate real-time challenge of bundling and matching passengers—each with distinct origins and destinations—to available vehicles, all while navigating significant system uncertainties. Due to the extensive observation space arising from the large number of drivers and orders, order dispatching, though fundamentally a centralized task, is often addressed using Multi-Agent Reinforcement Learning (MARL). However, independent MARL methods fail to capture global information and exhibit poor cooperation among workers, while Centralized Training Decentralized Execution (CTDE) MARL methods suffer from the Curse of Dimensionality (CoD). To overcome these challenges, we propose Triple-BERT, a centralized framework designed specifically for large-scale order dispatching on ride-sharing platforms based on Single Agent Reinforcement Learning (SARL). Built on a variant TD3, our approach addresses the vast action space through an action decomposition strategy that breaks down the joint action probability into individual driver action probabilities. To handle the extensive observation space, we introduce a novel BERT-based network, where parameter reuse mitigates parameter growth as the number of drivers and orders increases, and the attention mechanism effectively captures the complex relationships among the large pool of driver and orders. We validate our method using a real-world ride-hailing dataset from Manhattan. Triple-BERT achieves approximately an 11.95% improvement over current state-of-the-art methods, with a 4.26% increase in served orders and a 22.25% reduction in pickup times. Our code, trained model parameters, and processed data are publicly available at `https://github.com/RS2002/Triple-BERT`. [1]

## 1 Introduction

Ride-sharing platforms, such as Uber and Lyft, face the complex challenge of dynamically matching passengers with distinct origins and destinations to available vehicles in real time. This task must account for significant system uncertainties, including fluctuating demand, varying traffic conditions, and the availability of drivers. As the volume of concurrent ride requests increases, these platforms must efficiently allocate resources to minimize detours, reduce waiting times, and maximize customer satisfaction and platform revenue. However, the inherently large and dynamically changing action and observation spaces make this problem highly challenging for the operation of ride-sharing platforms.

Recently, Reinforcement Learning (RL) methods have shown great potential in addressing the order dispatching problem in ride-sharing platforms. Model-free RL, in particular, enables agents to autonomously learn optimal dispatching policies by interacting with the environment, without requiring complex system modeling. This approach allows platforms to optimize multiple objectives, including platform income, driver payments, and customer satisfaction. Despite these advantages, applying RL to large-scale order dispatching introduces significant challenges. The vast action and observation spaces, stemming from the large number of drivers and orders, make sufficient

---

*Corresponding Author: Sen Li (cesli@ust.hk)
[1]Do We Really Need MARL for Order Dispatch on Ride-Sharing Platforms? In summary: No, because we developed a SARL method that achieves better global planning. However, yes, our method still requires MARL for pre-training to establish a strong starting point for SARL.

exploration and efficient training difficult. Multi-Agent Reinforcement Learning (MARL) methods have been widely adopted to address these challenges by decomposing the problem into smaller subproblems for individual agents (drivers). Independent MARL methods, such as Independent Double DQN (IDDQN) [1; 29; 53] and Independent SAC (ISAC) [64], are computationally efficient but fail to capture global information and exhibit limited cooperation among agents. Graph Neural Networks (GNNs) have been introduced to enable the network to capture neighboring information for each agent, alleviating this issue to certain extent [23; 57]. Meanwhile, Centralized Training with Decentralized Execution (CTDE) methods, such as QMIX [16] and Coordinated Policy Optimization (CoPO) [54], struggle with the Curse of Dimensionality (CoD) when applied to large-scale scenarios with thousands of agents, resulting in slow convergence and suboptimal performance. (Due to page limitations, we provide a detailed review of ride-sharing methods and cooperative MARL approaches in Appendix F.)

To address these limitations, this paper proposes a centralized Single-Agent Reinforcement Learning (SARL) method, named Triple-BERT, tailored for large-scale order dispatching in ride-sharing platforms. Triple-BERT introduces an action decomposition method that simplifies the joint action probability into individual driver action probabilities, enabling each driver to make independent decisions while maintaining global coordination. The method leverages TD3 [12] for optimization, with modifications to the actor optimization process via policy gradient [46] to better suit the ride-sharing context. To handle the extensive observation space, we design a novel BERT-based [8] neural network architecture. This network employs bi-directional self-attention to effectively capture complex relationships between drivers and orders, while its parameter reuse mechanism prevents parameter explosion as the number of drivers and orders increases. Additionally, compared to MARL, SARL faces a unique challenge of sample scarcity, as the records of multiple agents are merged into a single training stream. To address this, we propose a two-stage training strategy, where feature extractors are pre-trained using a MARL approach to learn general embedding capabilities, followed by centralized fine-tuning. The main contributions of this paper can be summarized as follows:

- We introduce Triple-BERT, which is the first centralized framework for large-scale order dispatching on ride-sharing platforms based on a variant centralized TD3. This framework addresses the limitations of the observation space and the inefficiencies in cooperation among agents present in conventional MARL-based methods. To tackle the large action space inherent in the matching problem of order dispatching tasks, we propose an action decomposition method that breaks down the joint action probability into individual driver action probabilities. Additionally, we propose a two-stage training method to address the sample scarcity issue in SARL, where the feature extractors are first trained using a MARL approach.

- To support the proposed RL framework in a large observation space, we develop a novel neural network architecture based on BERT. This design leverages self-attention mechanisms to effectively capture the relationships between drivers and orders. Furthermore, we incorporate a QK-attention module to reduce computational complexity from multiplication to addition in the order dispatching task, along with a positive normalization method to mitigate parameter redundancy issues.

- We validate the proposed method in the ride sharing scenario, using a real-world dataset of ride-hailing trip records from Manhattan. Our method outperforms the MARL methods reported in previous works, demonstrating approximately a 11.95% improvement over current state-of-the-art methods, with a 4.26% increase in served orders and reductions of about 22.25% in pickup time.

## 2 PROBLEM SETUP

In this paper, we address the order dispatching task within on-demand ride sharing platform. We consider a platform managing $n$ drivers (hereafter referred to as workers), represented by the state $W_t = \{w_{1,t}, w_{2,t}, \ldots, w_{n,t}\}$, where $w_{i,t}$ denotes the state of worker $i$ at time $t$. At each time step, the platform processes a set of orders, including newly arrived orders and any previously unassigned orders, denoted as $O_t = \{o_{1,t}, o_{2,t}, \ldots, o_{m_t,t}\}$, where $m_t$ is the total number of orders at time $t$. Since real-time performance is crucial in on-demand systems, the platform aims to bundle and assign orders in a way that minimizes delivery time while maximizing the number of served orders. Customers are assumed to be impatient; if an order is not acknowledged within a specified time frame, customers will decline it. Moreover, late deliveries beyond the scheduled time may result in customer complaints, potentially causing losses for the platform. The overall workflow is illustrated

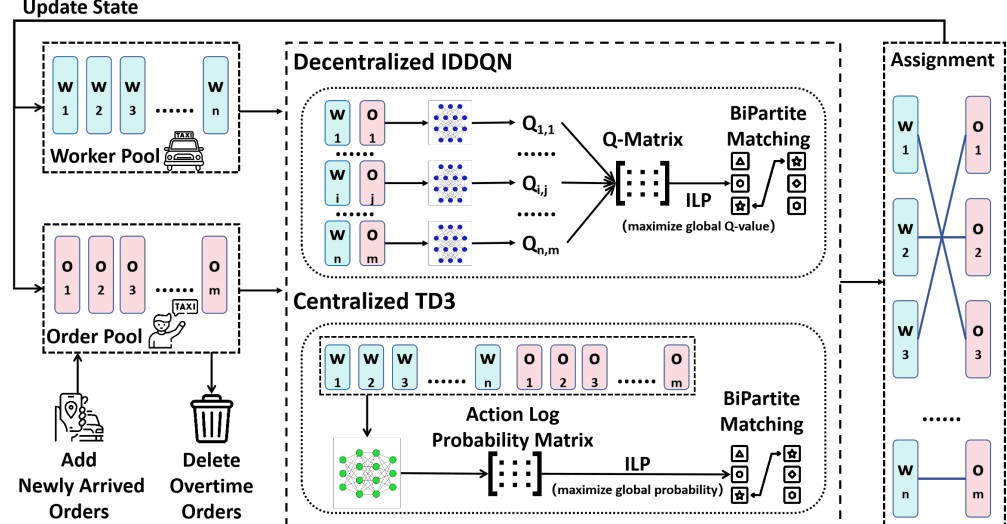

Figure 1: Workflow: At each time step, the worker and order pools update their states based on the assignments made in the previous time step. Specifically, the order pool adds newly arrived orders and removes overdue ones. For IDDQN, the Q-value of each worker-order pair is calculated, and ILP is applied to maximize the global Q-value. For TD3, the probability of each worker-order pair is computed, followed by the application of ILP to maximize the global assignment probability.

in Fig. 1, and the Markov Decision Process (MDP) is formulated as $< S, A, R, P >$, encompassing the state, action, reward, and transition function, which will be detailed below:

**(i) State:** At timestep $t$, the state or observation can be represented as $S_t = [W_t, O_t]$, consisting of the states of workers and orders. For the order $j$ to be assigned, the state $o_{i,j}$ includes the order's origin and destination, pickup time, and scheduled arrival time. For each worker $i$, the state $w_{i,t}$ consists of the onboard orders $H_{i,t}$ that are still unfinished, the current location, the residual capacity, and the estimated time when he/she will be available to accept a new order. (Note that we assume if a worker is en route to pick up a new order or if his/her capacity is full, he/she cannot serve a new order.) Specifically, $H_{i,t}$ is a sequence of orders $H_{i,t} = \{h_{i,1,t}, h_{i,2,t}, \ldots, h_{i,k_{i,t},t}\}$, where $k_{i,t}$ is the number of onboard orders for worker $i$ at time $t$ and each order $h_{i,k,t}$ contains the same information as the orders to be assigned $o_{j,t}$.

**(ii) Action:** At each time $t$, the action can be represented as $A_t = \{a_{1,t}, a_{2,t}, \ldots, a_{n,t}\}$, where each $a_{i,t}$ is an $m_t$-dimensional vector with at most one element set to 1, indicating which order is assigned to worker $i$. The order dispatching task is particularly challenging due to two main factors: (i) the size of the action space keeps changing over time because the number of orders $m_t$ varies dynamically as new orders arrive and old orders are completed or canceled; (ii) the size of the action space is extremely large for real systems. For instance, considering $n = 1000$ workers and $m_t = 10$ orders, the action space can reach approximately $10^{30}$. (A detailed proof is provided in Appendix A.) This combination of an enormous action space and its continuously changing size significantly complicates sufficient exploration and stable network convergence for standard RL methods.

**(iii) Reward Function:** We split the reward function for each worker, meaning each worker will receive a reward $r_{i,t+1}$ at time step $t$, and the global reward is the sum of each worker's reward: $R_{t+1} = \sum_{i=1}^{n} r_{i,t+1}$. The reward $r_{i,t+1}$ can be calculated according to the following function:

$$r_{i,t+1} = \mathcal{R}(s_{i,t}, a_{i,t}) = \begin{cases} \beta_1 + \beta_2 p_{i,t}^{in} - \beta_3 p_{i,t}^{out} - \beta_4 \chi_{i,t} - \beta_5 \rho_{i,t} , & |a_{i,t}| = 1 \\ 0 , & |a_{i,t}| = 0 \end{cases} \quad (1)$$

where $\beta_1$ to $\beta_5$ are non-negative weights representing the platform's valuation of each term, $p_{i,t}^{in}$ and $p_{i,t}^{out}$ represent the income from customers and the payout to workers, respectively. The variables $\chi_{i,t}$ and $\rho_{i,t}$ represent the number of en-route orders that will exceed their scheduled time and the additional travel time of all en-route orders when the assigned order is added to the scheduled route of worker $i$ at time $t$, respectively. This reward function is designed to comprehensively consider

the interests of the platform, workers, and customers, mimicking the operation of a real-world ride sharing platform. It is important to emphasize that $p_{i,t}^{in}$ and $p_{i,t}^{out}$ are calculated based on the order distance and the additional travel distance for the worker, respectively. When calculating travel time, we will utilize the Traveling Salesman Problem (TSP) to optimize the worker's route.

**(iv) Transition Function:** In our system, the reward is deterministic given the current state and action. Therefore, the transition function is represented by $P(S_{t+1}|S_t, A_t)$. In this study, the transition probabilities are not explicitly modeled; instead, they are inferred through the model-free RL.

## 3 METHODOLOGY

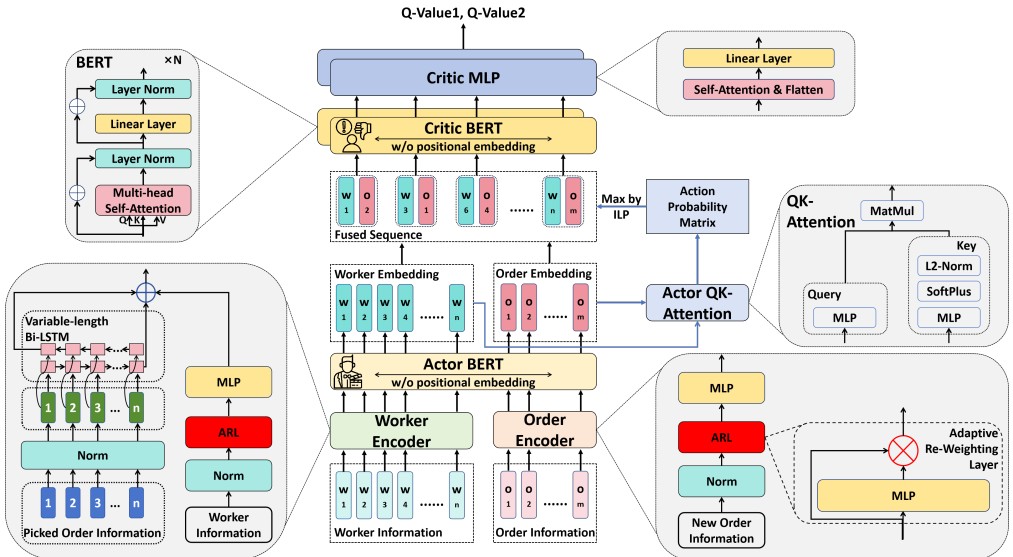

Figure 2: Network Architecture: The network consists of three main components: the feature extractor, the actor sub-network, and the critic sub-network. First, a worker encoder and an order encoder are used to extract features from individual worker and order information, respectively. Then an Actor BERT model captures the relationships between them and a QK-Attention module calculates the selection probabilities for each worker-order pair. Finally, the fused features of the selected worker-order pairs are input into two separate Critic BERT models for further information extraction, and two Critic MLPs compute the Q-values, as TD3 requires two critics. (In this figure, the fused sequence (input to Critic-BERT) represents workers 1, 3, 6, and $n$ selecting orders 2, 3, 4, and $m$, respectively.)

### 3.1 OVERVIEW

In this work, we aim to utilize centralized SARL to address the large-scale order dispatching task, with the goal of enabling the model to fully leverage global information to enhance cooperation among workers. To tackle the challenges of large action and observation spaces, we propose a novel network architecture, as illustrated in Fig. 2. This architecture employs the BERT model [8] to effectively extract the relationships between workers and orders using the self-attention mechanism. Additionally, an improved QK-attention [68] is implemented to reduce the computational complexity associated with the order dispatching task. Furthermore, we introduce an action decomposition method that breaks down the choice probability of each action within the vast action space into individual action probabilities for each worker selecting each order. Finally, to address the data scarcity challenge in MARL, we propose a two-stage training method, as shown in Fig. 1. In the first stage, we train the upstream layers of the network using the IDDQN approach, allowing them to develop general feature extraction capabilities. Subsequently, we train the entire neural network using centralized TD3 to realize better cooperation between workers.

## 3.2 NETWORK ARCHITECTURE

The proposed network structure is shown as Fig. 2, which constists of three parts: encoders (embed the worker and order information to a common feature space), actor sub-network (a BERT to extract the relationship between different workers and orders and a QK-Attention to generate the utility/probability of each worker-order pair), and critic sub-network (two BERT taking output of actor BERT as input and output the Q-value respectively).

### 3.2.1 FEATURE EXTRACTORS

At each time step, the network takes the entire state $S_t = [W_t, O_t]$ as input. We consider this as a combination of two sequences: $W_t$ and $O_t$. For each element $w_{i,t}$ and $o_{j,t}$, we employ two distinct encoders, referred to as the "Worker Encoder" and the "Order Encoder", to embed them separately into a feature space of the same dimension, allowing them to be input into a single BERT model.

Each worker state $w_{i,t}$ consists of two parts: an on-board order sequence and other non-sequence information. For the order sequence, a bi-directional LSTM [17] is utilized to extract its features. This approach effectively encodes variable-length sequences into a uniform dimensional feature space, addressing the CoD associated with conventional MLP encoders, where the number of parameters increases with sequence length. For non-sequence information, an MLP is employed for feature extraction. Finally, the two features are combined into a primary feature $\tilde{w}_{i,t}$. For the orders to be assigned $o_{j,t}$, an MLP is also used to extract the feature $\tilde{o}_{j,t}$. Notably, the dimensions of $\tilde{w}_{i,t}$ and $\tilde{o}_{j,t}$ are identical, and their information is concatenated into a sequence represented as $\tilde{S}_t = [\tilde{w}_{1,t}, \tilde{w}_{2,t}, \ldots, \tilde{w}_{n,t}, \tilde{o}_{1,t}, \tilde{o}_{2,t}, \ldots, \tilde{o}_{m_t,t}]$.

Additionally, to facilitate network convergence and enhance the extraction of input features, we incorporate a normalization layer and an Adaptive Re-weighting Layer (ARL) [4]. Given that different parts of the input may have varying magnitudes, which can impede model training, the normalization layer effectively addresses this issue. Furthermore, since different parts of the input carry different levels of importance, we utilize the ARL to enable the model to learn these variations, represented as: $y = x \circ \Omega$, where $x$ denotes the input, $\Omega$ represents the weight vector, calculated by $\Omega = \text{MLP}(x)$, and $\circ$ indicates the element-wise product.

### 3.2.2 ACTOR SUB-NETWORKS

The Actor sub-network consists of a BERT [8] model for feature extraction and a QK-attention module [68] for action decomposition and generation, which we will introduce in turn. In the feature extractors, we have already extracted the primary features from each worker and order state separately. To further explore the relationships between workers and orders, we utilize the BERT model, where the self-attention mechanism can effectively capture these relationships: $\overline{S}_t = [\overline{w}_{1,t}, \overline{w}_{2,t}, \ldots, \overline{w}_{n,t}, \overline{o}_{1,t}, \overline{o}_{2,t}, \ldots, \overline{o}_{m_t,t}] = \text{Actor-BERT}(\tilde{S}_t)$. Specifically, due to the permutation invariance of our input sequence, we omit the positional embedding in BERT, ensuring that the order in $S$ does not influence the encoding result. In contrast to conventional MARL methods like [23; 16], which encode each worker with its neighboring states to gain a broader perspective, our Actor-BERT directly aggregates global worker information, facilitating more effective cooperative dispatching between workers.

In conventional order dispatching tasks, the typical approach to address the dynamic action space (related to the number of orders) involves evaluating each worker-order pair separately and finding the optimal dispatching solution based on these evaluations. However, this approach has two significant shortcomings. First, it neglects the relationships between orders, which we address through the self-attention mechanism in BERT, capturing not only the relationships between workers but also between orders and between orders and workers. Second, evaluating each worker-order pair is time-consuming and resource-intensive: $\text{F}(\overline{w}_{i,t}, \overline{o}_{j,t}; \theta_F) \in \mathbb{R}^1$, where F is the network and $\theta_F$ represents its parameters. The complexity can be represented as $O(|\text{F}| \cdot n \cdot m_t)$, where $|\text{F}|$ denotes the complexity of the neural network. To mitigate this issue, we employ a QK-attention module [68], represented as:

$$\text{QK-Attention}(\overline{w}_{i,t}, \overline{o}_{j,t}) := \text{f}(\overline{w}_{i,t}; \theta_f) \cdot \text{g}(\overline{o}_{j,t}; \theta_g)^T \approx \text{F}(\overline{w}_{i,t}, \overline{o}_{j,t}; \theta_F), \qquad (2)$$

where f and g are two smaller networks, and $\theta_f$ and $\theta_g$ are their parameters. The intuition behind QK-attention is to use two smaller networks to approximate a larger network, similar to the motivation

behind LoRA [20]. In this way, the complexity of computing all worker-order pairs becomes $O(|f| \cdot n + |g| \cdot m_t + d \cdot n \cdot m_t)$, where $|f|$ and $|g|$ are the complexities of the two neural networks, $d$ is their output dimension, and $d \cdot n \cdot m_t$ is the complexity of matrix multiplication. Here, $d$ is very small, making $d \cdot n \cdot m_t$ much smaller than the neural network computation complexity, i.e., $d \cdot n \cdot m_t \ll |f| \approx |g| < |F|$. Thus, we have $O(|f| \cdot n + |g| \cdot m_t + d \cdot n \cdot m_t) < O(|F| \cdot (n + m_t)) < O(|F| \cdot n \cdot m_t)$, indicating that the QK-attention successfully transforms the multiplication complexity of evaluating each worker-order pair into addition complexity.

However, we observe a parameter redundancy issue in Equation 2, which can lead to potential instability during training. This redundancy arises because there are actually infinite solutions for $f$ and $g$, as $f' = \alpha f$ and $g' = \frac{g}{\alpha}$ is also a valid solution for any non-zero real vector $\alpha$. Inspired by Dueling DQN [58], we propose a positive normalization method:

$$\text{QK-Attention-Norm}(\overline{w}_{i,t}, \overline{o}_{j,t}) := \text{f}(\overline{w}_{i,t}; \theta_f) \cdot \frac{\text{Softplus}(\text{g}(\overline{o}_{j,t}; \theta_g))^T}{||\text{Softplus}(\text{g}(\overline{o}_{j,t}; \theta_g))||_2} \ . \tag{3}$$

This normalization ensures that the elements in $\frac{\text{Softplus}(\text{g}(\overline{o}_{j,t}; \theta_g)^T)}{||\text{Softplus}(\text{g}(\overline{o}_{j,t}; \theta_g)^T)||_2}$ are always non-negative, with an L2 norm of 1. Although this approach does not guarantee a unique solution, it significantly improves training stability, as demonstrated by our experimental results in Section 4. In our task, the output of the QK-attention is a matrix $M_t \in \mathbb{R}^{n, m_t}$, representing the utility of each worker choosing each order, which will be detailed in Section 3.3.2.

### 3.2.3 CRITIC SUB-NETWORKS

The role of the critic is to evaluate the quality of actions, with the detailed action generation method introduced in Section 3.3.2. We first define an action function $\mathcal{A}$:

$$\mathcal{A}(w_{i,t}) = \begin{cases} (\overline{w}_{i,t}, \overline{o}_{j,t}) & \text{if order } j \text{ is assigned to worker } i \text{ at time } t \\ \emptyset & \text{if no order is assigned to worker } i \text{ at time } t \end{cases} \tag{4}$$

where $\overline{w}_{i,t}$ and $\overline{o}_{j,t}$ are the outputs of Actor-BERT, and $(\overline{w}_{i,t}, \overline{o}_{j,t})$ represents the combination of the two vectors into a single feature vector. We then construct a new sequence: $\dot{S}_t = [\mathcal{A}(w_{1,t}), \mathcal{A}(w_{2,t}), \dots, \mathcal{A}(w_{i,t})]$. Another BERT network, referred to as "Critic-BERT", is used to further extract features from $\dot{S}_t$, represented as $\ddot{S}_t = \text{Critic-BERT}(\dot{S}_t)$. A self-attention mechanism and a linear layer (collectively named Critic-MLP) are then utilized to estimate the Q-value from $\ddot{S}_t$ (for detailed processing methods, refer to [6]). Furthermore, as TD3 [12] requires two critics, we employ two distinct Critic-BERT and Critic-MLP networks. These share the input features from Actor-BERT but process them separately.

### 3.3 TRAINING PROCESS

### 3.3.1 STAGE 1: DECENTRALIZED IDDQN TRAINING

In this stage, we aim to first train the feature-extracting capacity of the worker encoder and order encoder using a substantial number of samples. To obtain sufficient samples, we view the dispatching problem as a multi-agent scenario, where at each time step, each agent can access its own record. We adopt the independent assumption that all agents share the same policy, allowing for the sharing of records between agents and leading to a large experience replay buffer.

Since our goal in this stage is not to train a powerful model but rather to enable the feature extractor to learn its general feature-extracting capabilities, we select the simplest yet efficient method for order dispatching, namely, the IDDQN. Each worker is treated as an independent agent with the state defined as $s_{i,t} = [w_{i,t}, O_t]$ at time $t$. We employ a neural network to estimate the Q-value at each step as $Q^{DQN}_{\pi^Q_\Phi}(s_{i,t}, a_{i,t})$, where $\Phi$ represents the network parameters and $\pi^Q_\Phi$ denotes the strategy.

To construct the network, we utilize QK-attention to process the outputs of the worker encoder and order encoders to estimate the Q-value for each worker-order pair, represented as QK-Attention-Norm($\tilde{w}_{i,t}, \tilde{o}_{j,t}$) (denoted as $y_{i,j,t}$). Although the state space encompasses the entire

order state from $o_{1,t}$ to $o_{m_t,t}$, we focus on a single order $o_{j,t}$ when computing the Q-value for choosing order $j$. This approach aligns with previous work such as [23; 21], as the entire order state can be excessively large for a simple network to learn (our Triple-BERT effectively addresses this issue) and many networks struggle to process variable dimensional inputs (with order amounts varying at each time step). Consequently, we can compute a Q-matrix $Y_t \in \mathbb{R}^{n,m_t}$, where the element in the $i$-th row and $j$-th column, $y_{i,j,t}$, represents the Q-value of assigning order $j$ to worker $i$ at time $t$. The core strategy of IDDQN is to maximize the global Q-value, expressed as $Q(S_t, A_t) = \sum_{i=1}^{n} Q(s_{i,t}, a_{i,t})$ at each time step. To achieve this, we construct a bipartite graph where each worker and order is represented as a node. An arbitrary worker $i$ and order $j$ are linked by an edge weighted by the Q-value of this worker selecting this order at the current time, i.e., $y_{i,j,t}$. We then utilize Integer Linear Programming (ILP) to solve this maximizing bipartite matching problem. (To avoid assigning orders to unavailable workers—those at full capacity or on their way to pick up an assigned order—we set the Q-value of all actions for such workers in the Q-matrix $Y_t$ to $-\infty$.) A detailed construction of the problem is provided in Appendix B.1. For the training of IDDQN, it follows the same process of previous work [23]. Due to page limitation, we detailed it in Appendix D.1.

*Here, We want to make some explanations about the independent assumption. We acknowledge that the independent assumption can be unreliable, which indeed hinders the performance of conventional independent MARL-based methods. However, previous works [40; 10; 56] have shown the efficacy and simplicity of independent MARL methods. As a result, even if the performance of independent MARL is not satisfactory, it does provide a good starting point for our centralized SARL method. In our approach, the independent assumption is only utilized during the pre-training stage to warm up the model. After pre-training, our centralized SARL framework no longer relies on the independence assumption.*

### 3.3.2 STAGE 2: CENTRALIZED TD3 TRAINING

In the standard AC framework, the process can be summarized as follows: an actor network generates actions based on the current state, represented as $A_t = \text{Actor}(S_t; \theta_A)$, while a critic network evaluates these actions using $\hat{Q}_t = \text{Critic}(S_t, A_t; \theta_C, \pi_{\theta_A}^T)$. Here, $\theta_A$ and $\theta_C$ are the parameters of the actor and critic networks, respectively, and $\pi_{\theta_A}^A$ denotes the strategy of AC. During training, the critic network is updated using TD-error, similar to Q-learning, and the actor network is updated to maximize $\hat{Q}$. However, a challenge mentioned in Section 2 is that the action space is too large for the order dispatching scenario. Additionally, the actions in order dispatching are discrete, complicating optimization using TD3. To address these issues, we propose an action decomposition method along with a policy gradient-style optimization method.

Before delving into the details, we denote both $\theta_A$ and $\theta_C$ with the parameters $\Theta$, as in our network (Fig. 2), the actor and critic share the same architecture. The trained network parameters from Stage 1, $\Phi$, are part of $\Theta$. Moreover, the policy of TD3 is represented as $\pi_\Theta^T$.

**(i) Actor:** In the standard AC framework for discrete action problems, the policy network generates probabilities for each action, from which actions are sampled. However, in the ride-sharing task, this approach encounters two significant challenges: (i) First, the action space is exceedingly large. As shown in Appendix A, a typical scenario with 1,000 couriers and 10 orders can yield nearly $10^{30}$ possible actions. (ii) Second, because the orders vary at each step (including both order volume and content), it is impossible to generate a fixed-dimension action probability vector as is customary in the standard AC framework. (iii) Third, due to the dependency among drivers (an order cannot be assigned to multiple workers concurrently), treating drivers as independent individuals for separate action sampling is impractical.

To address these issues, we impose structural assumptions on the policy function to facilitate its derivation (i.e. the proposed action decomposition strategy). We define $\mathscr{P}_{i,j,t}$ as the probability that worker $i$ chooses order $j$ at time $t$, based on the logit model [33]. Specifically, we first expand the utility matrix $M_t$ generated by the Actor QK-Attention to $\mathcal{M}_t = [M_t, N_t] \in \mathbb{R}^{n,m_t+1}$, where $N_t$ is an $n$-dimensional vector representing the utility of each worker choosing no order. This vector is obtained by processing the output of Actor-BERT with a MLP, expressed as $N_t = \text{MLP}([\overline{w}_{1,t}, \overline{w}_{2,t}, \ldots, \overline{w}_{n,t}])$. This allows us to compute the probability of each worker choosing each action using the logit model, yielding $\mathscr{P}_t = \text{Softmax}(\mathcal{M}_t)$.

Furthermore, we assume that the aggregate policy is derived from the product of these probabilities:

$$\pi_\Theta^T(A_t|S_t) = \text{z} \left( \prod_{i,j \in \text{h}(A_t)} \mathscr{P}_{i,j,t} \right), \tag{5}$$

where $\text{z}(\cdot)$ is an increasing function that also depends on the current state $S_t$ (which we omit for simplicity), and $\text{h}()$ is defined as $\text{h}(A_t) = \{(i,j)|a_{i,j,t} = 1\}$. This is a reasonable simplification, as it implies that if an action $A_t$ has a higher value of $\prod_{i,j \in \text{h}(A_t)} \mathscr{P}_{i,j,t}$, it has a greater probability of being chosen. *We wish to emphasize that the probability $\mathscr{P}$ does not exist in reality but serves as a virtual construct defined by us. We connect the output of the network $\mathcal{M}_t$ to the policy $\pi_\Theta^T$ through this defined probability and the mapping function $\text{z}(\cdot)$. Essentially, we restrict the policy space to a smaller class as defined by Eq. 5 to facilitate optimization and application, as follows.*

However, defining and computing such a function $\text{z}(\cdot)$ is challenging due to the vast action space, complicating the sampling of an action from the strategy $\pi_\Theta^T(A_t|S_t)$. We define an efficient approach to address this. First, during inference, we can greedily select the action with the maximum probability, as this action should theoretically have the highest utility:

$$\arg \max_{A_t \in \psi(S_t)} \pi_\Theta^T(A_t|S_t) = \arg \max_{A_t \in \psi(S_t)} \text{z} \left( \prod_{i,j \in \text{h}(A_t)} \mathscr{P}_{i,j,t} \right) = \arg \max_{A_t \in \psi(S_t)} \sum_{i,j \in \text{h}(A_t)} \log \mathscr{P}_{i,j,t}, \tag{6}$$

where $\psi(S_t)$ is the set of all possible actions under the current state $S_t$. This holds because both $\text{z}(\cdot)$ and $\log(\cdot)$ are increasing functions. We can construct a bipartite graph similar to Stage 1, where each available worker and order is represented as a node, and the link between each worker $i$ and order $j$ at time $t$ is weighted by their log probability $\log \mathscr{P}_{i,j,t}$. By utilizing ILP, we can find the action $A_t$ that maximizes $\pi_\Theta^T(A_t|S_t)$. The bipartite graph construction process is detailed in Appendix B.2. During training, we introduce random noise to the probability matrix $\mathscr{P}_t$ and the model selects actions using the same method as in Eq. 6. When the noise is sufficiently large, the policy degrades to a totally random policy, and when the noise is zero, the policy converges to a greedy strategy. Although we cannot directly express the function $\text{z}(\cdot)$, it must ensure that the function is a increasing function (since the noise is totally random). More details about the noise can be found at Appendix C.

Optimizing this probability using vanilla TD3 is challenging due to the variable action space and the gap between action probabilities and the selected action (the gradient cannot propagate through them). To address this, we employ an approximate policy gradient optimization method [46]:

$$\nabla_\Theta \text{J}(\Theta) \propto \mathbb{E}_{\pi_\Theta^T} \left[ (\text{Q}_{\pi_\Theta^T}^{TD3}(S_t, A_t) - B) \nabla_\Theta \sum_{i,j \in \text{h}(A_t)} \log \mathscr{P}_{i,j,t} \right], \tag{7}$$

where $\text{J}(\Theta)$ is the optimization objective (long-term cumulative reward), $B$ is a baseline independent of state (we simplify by setting it to 0), and $\text{Q}_{\pi_\Theta^T}^{TD3}(S_t, A_t)$ is the Q-value under the policy $\pi_\Theta^T$, which can be estimated by $\text{Q}_{\pi_\Theta^T, i}^{TD3}(S_t, A_t; \Theta)$ using our proposed network ($i = 1, 2$, as there are two estimated Q-values in TD3). Detailed derivations can be found in Appendix C. We then use gradient ascent to maximize $\text{J}(\Theta)$, thus the loss function for the actor can be expressed as $L_A = -\nabla \text{J}(\Theta)$.

**(ii) Critic:** For the critic, it can be updated in a manner similar to vanilla TD3, where the loss function can be expressed as:

$$L_C = \sum_{i=1,2} \mathbb{E}_{\pi_\Theta^T} \left[ \mathcal{Q}_{\pi_{\Theta^-}^T}^{TD3}(S_{t+1}, R_{t+1}; \Theta^-) - \text{Q}_{\pi_\Theta^T, i}^{TD3}(S_t, A_t; \Theta) \right],$$
$$\mathcal{Q}_{\pi_{\Theta^-}^T}^{TD3}(S_{t+1}, R_{t+1}; \Theta^-) = R_{t+1} + \gamma \min_{i=1,2} \text{Q}_{\pi_{\Theta^-}^T, i}^{TD3}(S_{t+1}, \text{Actor}(S_{t+1}; \Theta^-, \xi); \Theta^-), \tag{8}$$

where $\mathcal{Q}_{\pi_{\Theta^-}^T}^{TD3}$ is the learning target function, $\Theta^-$ represents the parameters of the target network, which updates more slowly than the policy network $\Theta$ to provide a stable target, and $\xi$ is a small random noise applied in the probability matrix $\mathscr{P}$. More details can be viewed in Appendix D.2.

Table 1: Comparison of Different Ride Sharing Methods: **Bold** entries represent the best results.

| Method | DeepPool [1] | BMG-Q [23] | HIVES [16] | Enders et al. [9] | CEVD [3] | Triple-BERT |
|---|---|---|---|---|---|---|
| **Type** | Independent | | CTDE | | Centralized | |
| **RL Algorithm** | IDDQN [48] | IDDQN [48] | QMIX [38] | MASAC [14] | VD$^2$ [45] | IDDQN [48] $\rightarrow$ TD3 [12] |
| **Multi-Agent** | ✓ | ✓ | ✓ | ✓ | ✓ | ✓ $\rightarrow$ ✗ |
| **Reward Type** | Local | Local | Global | Local | Global | Local $\rightarrow$ Global |
| **Network Backbone** | MLP$^3$ | GAT [51] | GRU [5] | MLP+Attention | MLP | BERT [8] |
| **Model Size** | **20K** | 117K | 16M | 118K | 23K | 16M |
| **GPU Occupation (GB)** | **3.97** | 4.28 | 6.01 | 8.19 | 21.45 | 8.03 |
| **Average Reward** ($10^3$) | 12.72 | 13.04 | 12.37 | 12.04 | 13.16 | **14.73** |

## 4 EXPERIMENT

To validate the proposed method, we evaluate its performance in the ride sharing dispatching task using real-world yellow ride-hailing data from Manhattan, New York City [47][4] . To illustrate the efficiency and superiority of our proposed Triple-BERT, we compare it with several previous ride sharing methods of different types, including Independent MARL, CTDE MARL, and Centralized MARL, as shown in Table 1. Detailed information regarding our experiment configuration, simulator setup, and a comprehensive description of the comparative experiment can be found in Appendix E.

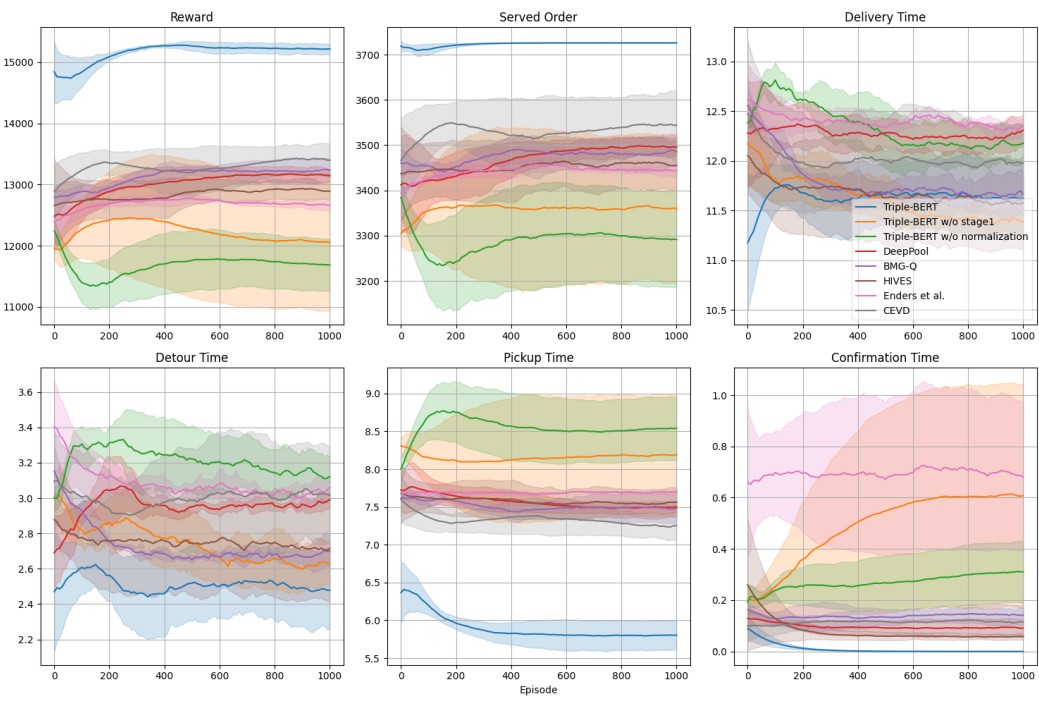

Figure 3: Training Process: Each method is trained five times, and the curve is smoothed using Exponential Moving Average (EMA) with $\alpha = 0.1$. The shaded area represents the standard deviation.

As shown in Fig. 3, we first illustrate the training process of different models by evaluating their performance in the training scenario every 10 episodes. The six sub-figures depict the cumulative reward, the number of orders served, and the average delivery time, detour time, pickup time, and confirmation time for each order. It is evident that our method outperforms the other models in most

---

[2]The original VD is a CTDE method. However, the CEVD variant modifies it to a centralized version.

[3]In the original paper fo DeepPool, the authors used CNN. However, due to differences in the observation space of our task, we replaced it with MLP.

[4]https://www.nyc.gov/site/tlc/about/tlc-trip-record-data.page

metrics, with the cumulative reward exceeding that of the best alternative method by approximately 15%. The highest number of served orders indicates that our method achieves better cooperation among workers. We then evaluate these methods over different periods, and the average rewards are shown in Table 1, where our method also demonstrates the best performance. More details about the experimental results can be found in Appendix E.4.

To further illustrate the generalization capacity of our method, we evaluate it in additional trials, specifically utilizing all-day data from July 18, 2024, drawn from High Volume For-Hire Vehicle (FHV) trip data [47]. This dataset features a different order distribution compared to the yellow-taxi data used for training. The 24-hour data generates 48 episodes, each lasting 30 minutes, during which the order volume varies from 734 to 5,989 in each episode. The experimental results are presented in Table 2. The findings remain consistent with previous results: our Triple-BERT model achieves the highest reward by optimizing pickup time to improve the service rate, even if this leads to higher delivery and detour times due to increased order bundling. Additionally, we observe that our method exhibits a higher standard deviation in rewards compared to others. This is because, in low-order volume scenarios, performance among different methods does not substantially differ, as all orders can be effectively served. However, in high-order volume scenarios, our Triple-BERT model demonstrates a significant advantage, resulting in higher rewards. Consequently, this contributes to a greater standard deviation when aggregating all scenarios. More expanded experiment could be found at Appendix E.

Table 2: Performance of Different Methods in Manhattan FHV Data [47]

| Method | Reward | Service Rate | Delivery Time | Detour Time | Pickup Time | Confirmation Time |
|---|---|---|---|---|---|---|
| **DeepPool** [1] | 11258.56±2578.68 | 0.76±0.18 | 13.93±0.93 | 2.54±1.48 | 10.19±0.78 | 0.09±0.07 |
| **BMG-Q** [23] | 11899.39±2804.65 | 0.78±0.17 | 13.27±1.00 | 2.44±1.54 | 9.39±0.43 | 0.15±0.11 |
| **HIVES** [16] | 11183.12±2458.29 | 0.78±0.18 | 13.72±1.48 | 2.87±1.90 | 9.77±0.68 | **0.05±0.03** |
| **Enders et al.** [9] | 10512.65±2744.34 | 0.78±0.17 | 14.12±0.48 | 3.06±0.43 | 9.92±0.55 | 1.37±0.99 |
| **CEVD** [3] | 12556.74±3303.93 | 0.80±0.13 | **12.33±0.72** | **2.25±1.33** | 8.02±1.27 | 0.09±0.08 |
| **Triple-BERT** | **14329.74±4627.26** | **0.88±0.11** | 13.07±0.61 | 2.78±0.92 | **7.02±0.88** | 0.34±0.32 |

Then to demonstrate the model's efficiency, we conduct a series of ablation studies. In terms of model training, we compare the performance of the model with and without stage 1 pre-training. Regarding the network structure, we primarily compare the QK-Attention mechanism with and without the proposed positive normalization module. The detailed results are shown in Fig. 3. We observe that without stage 1 pre-training, the model fails to converge and exhibits significant fluctuations. Particularly in the later stages, the reward begins to decrease, which can be attributed to the lack of samples. Additionally, without the proposed normalization in QK-Attention, the model performs poorly, underperforming compared to all other methods. This is due to parameter redundancy, which leads to substantial fluctuations and hinders efficient learning.

## 5 CONCLUSION

In this work, we propose the first centralized SARL method, Triple-BERT, for large-scale order dispatching in ride-hailing platforms. Our method successfully addresses the challenge of large action spaces through an action decomposition technique and tackles the issue of sample scarcity with a proposed two-stage training method. The novel network also addresses the large observation space challenge by leveraging the self-attention mechanism of BERT. Additionally, we introduce an improved QK-Attention mechanism to reduce the computational complexity of order dispatching. Through experiments on real-world ride sharing data, we demonstrate that our method significantly outperforms conventional MARL methods, achieving better cooperation among drivers.

However, compared to traditional MARL-based ride-sharing methods, Triple-BERT is more sensitive to single points of failure, as its decisions depend on comprehensive information from all drivers and orders. Efficient strategies to address this dilemma warrant exploration in future work. Additionally, while this study represents the first centralized SARL-based approach to ride-sharing, we view it as merely the starting point for this new paradigm. Future research could focus on identifying more efficient SARL frameworks or enhancing our existing method, such as exploring importance sampling within our off-policy policy gradient-based actor optimization method, or investigating the use of offline training to replace our pre-training phase.

## ETHICS STATEMENT

*This work adheres to the principles outlined in the ICLR Code of Ethics.*

Efficient ride-sharing plays a crucial role in promoting convenient and sustainable urban transportation services. By enabling greater sharing among passengers, our method not only increases platform profitability and operational efficiency but also helps reduce total vehicle miles traveled and per-capita carbon emissions compared to solo rides. This, in turn, supports environmental sustainability goals. Moreover, our centralized reinforcement learning framework improves coordination among drivers, reduces delivery and detour times, and enables the platform to serve more orders within the same time frame. As a result, both platform income and customer satisfaction are enhanced, while also contributing to a greener and more efficient transportation system. We believe these contributions highlight the practical significance and societal value of our research.

However, the issue of algorithmic discrimination has received widespread attention over time. Closed-box management algorithms, including those for order dispatching, have been shown to create discriminatory scenarios for workers, as reinforcement learning methods primarily aim to maximize rewards without considering ethical implications. For example, algorithms may set different payment structures or order assignment preferences based on individual features or geographical locations of workers. We hope that our method will not exacerbate these issues and can be further developed to include constraints that promote fairness. Our goal is to strike a balance between profit and ethics, fostering a win-win situation for platforms, workers, and customers.

## REPRODUCIBILITY STATEMENT

The source code, trained parameters, and processed dataset are available at `https://github.com/RS2002/Triple-BERT`. The appendix also includes a detailed description of our methodology.

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

APPENDIX CONTENTS

## A  ACTION SPACE SIZE

The action space in our order dispatching task is given by:

$$|A_t| = \sum_{k=0}^{m_t} \mathrm{C}(m_t, k)\mathcal{P}(n, k) = \sum_{k=0}^{m_t} \frac{m_t!}{k!(m_t - k)!} \frac{n!}{(n - k)!} \,, \tag{9}$$

where $\mathcal{P}(n, k)$ represents the permutations of assigning $k$ orders to $n$ workers and $\mathrm{C}(m_t, k)$ represents the combinations of selecting $k$ orders from the total $m_t$ orders. This equation is based on two assumptions: (i) the platform will assign an arbitrary number of orders at each step (some orders yielding negative income will be declined by the platform) and (ii) the number of orders $m_t$ is less than the number of workers $n$, which can always be satisfied since $m_t$ represents the order count at only one timestep. Then we can derive the lower bound of $|A_t|$ as:

$$
\begin{aligned}
|A_t| = \sum_{k=0}^{m_t} \mathrm{C}(m_t, k)\frac{n!}{(n - k)!} &\geq \sum_{k=0}^{m_t} \mathrm{C}(m_t, k)(n - k + 1)^k \\
&\geq \sum_{k=0}^{m_t} \mathrm{C}(m_t, k)(n - m_t + 1)^k = (n - m_t + 2)^{m_t} \geq 2^{m_t} \quad (n \geq m_t \geq 0) \,.
\end{aligned}
\tag{10}
$$

As a result, the action space has a lower bound with the exponent to $m_t$. Consider the example in Section 2 where the number of workers $n$ is 1000 and the number of orders $m_t$ is 10. In this case, the expression $(n - m_t + 2)^{m_t}$ evaluates to $992^{10} \approx 10^{30}$.

## B  BIPARITE GRAPH CONSTRUCTION

### B.1  IDDQN BIPARTITE GRAPH

The bipartite graph in the IDDQN-based order dispatching method is constructed as follows:

$$\max_{A_t} \sum_{i \in \mathcal{I}} a_{i,j,t} \cdot y_{i,j,t}, \tag{11a}$$

$$\text{s.t.} \quad \sum_{i \in \mathcal{I}} a_{i,j,t} \leq 1, \quad \forall j \in \mathcal{J}_t, \tag{11b}$$

$$\sum_{j \in \mathcal{J}_t} a_{i,j,t} \leq 1, \quad \forall i \in \mathcal{I}, \tag{11c}$$

$$a_{i,j,t} \in \{0,1\}, \quad \forall i \in \mathcal{I}, j \in \mathcal{J}_t, \tag{11d}$$

where $a_{i,j,t}$ is the action representing whether worker $i$ is assigned order $j$ at time $t$ (with 1 indicating assignment and 0 indicating no assignment), $y_{i,j,t}$ denotes the Q-value of worker $i$ choosing order $j$ at time $t$ (with $y_{i,j,t} = -\infty$ for all unavailable workers at time $t$), $\mathcal{I}$ is defined as $\{1, 2, \ldots, n\}$, and the set $\mathcal{J}_t$ is defined as $\{1, 2, \ldots, m_t\}$. Constraint 11b ensures that an order can be assigned to at most one worker, while constraint 11c guarantees that each worker is assigned at most one order.

### B.2  TD3 BIPARTITE GRAPH

The bipartite graph in our proposed TD3-based order dispatching method is constructed as follows:

$$\max_{X_t} \sum_{i \in \mathcal{I}_t^w} x_{i,j,t} \cdot \log \mathscr{P}_{i,j,t}, \tag{12a}$$

$$\text{s.t.} \quad \sum_{i \in \mathcal{I}} x_{i,j,t} \leq 1, \quad \forall j \in \mathcal{J}_t, \tag{12b}$$

$$\sum_{j \in \mathcal{J}_t} x_{i,j,t} = 1, \quad \forall i \in \mathcal{I}_t^w, \tag{12c}$$

$$x_{i,j,t} \in \{0,1\}, \quad \forall i \in \mathcal{I}, j \in \mathcal{J}_t \cup \{m_t + 1\}, \tag{12d}$$

where $\mathcal{I}_t^w$ represents the set of available workers at time $t$. Here, constraint 12b does not apply in the last column, as it represents declining all orders, an action that can be chosen by any worker. Constraint 12c requires each row to equal 1, ensuring that each worker must either take an order or reject all, without other choices. We can then convert $X_t$ to action $A_t$ as follows:

$$a_{i,t} = \begin{cases} x_{i,j,t} & \text{if } i \in \mathcal{I}_t^w \text{ and } x_{i,m_t+1,t} = 0 \\ \mathbf{0} & \text{otherwise} \end{cases} \tag{13}$$

## C  POLICY GRADIENT PROOF

According to the policy gradient theory [46], we have:

$$
\begin{aligned}
&\nabla_\Theta \mathrm{J}(\Theta) \\
&\propto \mathbb{E}_{\pi_\Theta^T}\left[\left(\mathrm{Q}_{\pi_\Theta^T}^{TD3}(S_t, A_t) - B\right)\nabla_\Theta \log \pi_\Theta^T(A_t|S_t)\right] \\
&= \mathbb{E}_{\pi_\Theta^T}\left[\left(\mathrm{Q}_{\pi_\Theta^T}^{TD3}(S_t, A_t) - B\right)\nabla_\Theta \log \mathrm{z}\left(\prod_{i,j\in\mathrm{h}(A_t)}\mathscr{P}_{i,j,t}\right)\right] \\
&= \mathbb{E}_{\pi_\Theta^T}\left[\left(\mathrm{Q}_{\pi_\Theta^T}^{TD3}(S_t, A_t) - B\right)\frac{d\mathrm{z}(\prod_{i,j\in\mathrm{h}(A_t)}\mathscr{P}_{i,j,t})}{d\prod_{i,j\in\mathrm{h}(A_t)}\mathscr{P}_{i,j,t}}\frac{\prod_{i,j\in\mathrm{h}(A_t)}\mathscr{P}_{i,j,t}}{\mathrm{z}(\prod_{i,j\in\mathrm{h}(A_t)}\mathscr{P}_{i,j,t})}\nabla_\Theta \log \prod_{i,j\in\mathrm{h}(A_t)}\mathscr{P}_{i,j,t}\right] \\
&= \mathbb{E}_{\pi_\Theta^T}\left[\left(\mathrm{Q}_{\pi_\Theta^T}^{TD3}(S_t, A_t) - B\right)\mathcal{E}_{\mathrm{z}(x),x}|_{x=\prod_{i,j\in\mathrm{h}(A_t)}\mathscr{P}_{i,j,t}}\nabla_\Theta \sum_{i,j\in\mathrm{h}(A_t)}\log \mathscr{P}_{i,j,t}\right],
\end{aligned}
\tag{14}
$$

where $\mathcal{E}$ denotes elasticity, which measures the sensitivity of one variable to changes in another, and is defined as:

$$\mathcal{E}_{y,x} = \frac{d\log y}{d\log x} = \frac{dy}{dx}\frac{x}{y}. \tag{15}$$

As introduced in Section 3.3.2, the probability $\mathscr{P}$ and the mapping function $\mathrm{z}(\cdot)$ do not exist in reality but serve as virtual constructs used to connect the output to the policy $\pi_\Theta^T$ for simplified optimization and application. This means we can define them in any format we choose, and they restrict the policy space to a structural class as defined by Eq. 5.

To further simplify optimization, we define the format of $\mathrm{z}(\cdot)$ as $z(x) = ax^b$, where $a, b > 0$. This formulation has the advantage that the elasticity of $\mathrm{z}(\cdot)$ is a positive constant, i.e. $\mathcal{E}_{\mathrm{z}(x),x} = ab$. Thus, we have: $\nabla_\Theta \mathrm{J}(\Theta) \propto \mathbb{E}_{\pi_\Theta^T}\left[\left(\mathrm{Q}_{\pi_\Theta^T}^{TD3}(S_t, A_t) - B\right)\nabla_\Theta \sum_{i,j\in\mathrm{h}(A_t)}\log \mathscr{P}_{i,j,t}\right]$, corresponding to Eq. 7. This format of $\mathrm{z}(\cdot)$ aligns with our assumption that $\mathrm{z}(\cdot)$ should be an increasing function, which implies that if a driver-order pair has a higher probability of being chosen, it should also have a higher utility, resulting in a greater likelihood of being selected in the joint action.

As mentioned in Section 3.3.2, during training, we add random noise to $\mathscr{P}_t$ and then choose the action that maximizes $\sum_{i,j\in\mathrm{h}(A_t)}\log \mathscr{P}_{i,j,t}$. Currently, the mapping from $\sum_{i,j\in\mathrm{h}(A_t)}\log \mathscr{P}_{i,j,t}$ to the choosing probability $\pi_\Theta^T$ corresponds to $\mathrm{z}(\cdot)$. To further illustrate the robustness of our method, we compare the performance of our model using Gaussian noise [18], uniform noise, and binary symmetric channel (BSC) noise, where the noise follows a Bernoulli distribution and has been widely utilized in previous work [23; 21]. During training, we gradually reduce the noise to make the policy more deterministic. The experimental results are shown in Fig. 4, where we observe that, despite certain performance differences between the various types of noise, they all outperform conventional MARL methods. This suggests the efficiency and high robustness of our proposed method, indicating that the detailed expression of $\mathrm{z}(x)$ does not significantly influence the validation of the method based on Eq. 7, even if it may cause some performance gaps. The optimal noise for our task may require further exploration. *For fairness, we choose to use BSC noise when comparing with other methods, even though it appears to perform the worst among the three types of noise. We aim to demonstrate that our results are robust and superior, not relying on a particular choice of hyper-parameters or experiment scenarios.*

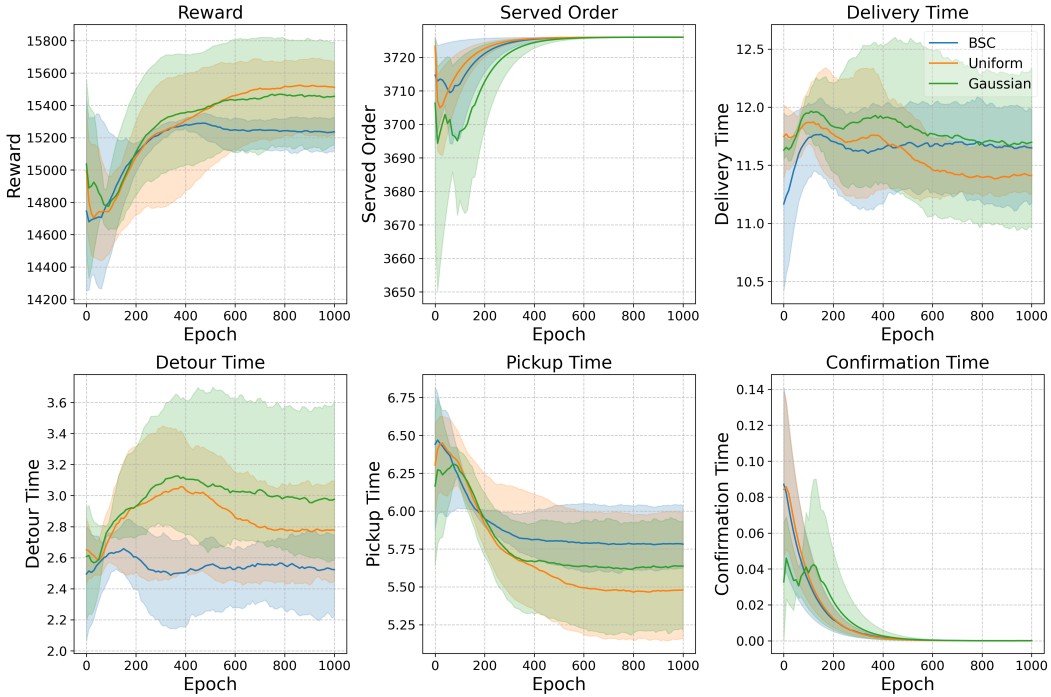

Figure 4: Comparison Between Different Noise Methods: Each method is trained three times, and the curve is smoothed using EMA with $\alpha = 0.1$. The shaded area represents the range of fluctuations, while the solid line indicates the average value.

# D  TRAINING PROCESS

## D.1  STAGE 1: IDDQN ALGORITHM

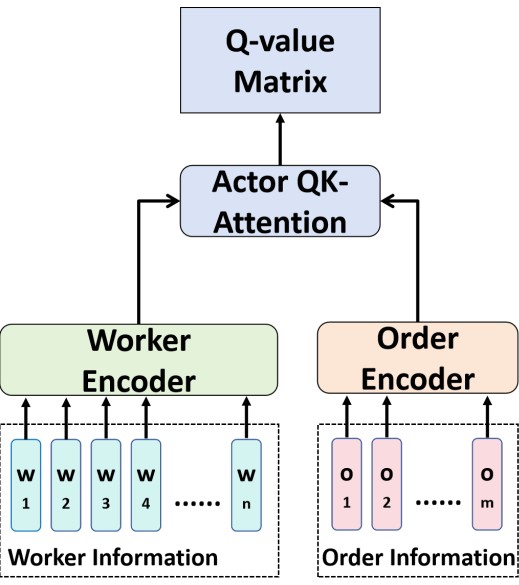

Figure 5: Network Structure in Stage 1

In stage 1, the network structure is shown as Fig. 5, which is consisted by the encoders and the QK-Attention module of proposed network in Fig. 2. *Although the model takes the entire worker and*

*order sequence as input, it primarily aims to utilize parallel computation to enhance computational efficiency. In the encoders, each worker and order's information is processed separately. Similarly, in the QK-Attention module, the Q-value for each worker-order pair is computed independently. It is also feasible to input only a single worker-order pair into this network, computing the Q-value exclusively for that pair; however, this would increase the computation time.*

During IDDQN training, we need to introduce some noise into the Q-matrix $Y_t$ to facilitate sufficient exploration. Specifically, for the $\epsilon$-greedy strategy, we randomly select a proportion $\epsilon$ of non-$-\infty$ elements in $Y_t$ and set them to a large positive number $\overline{Y}$ to enhance their likelihood of being selected. We then update the neural network by minimizing the TD-error, expressed as:

$$L_Q = \mathbb{E}_{\pi_\Phi^Q} \left[ \mathcal{Q}_{\pi_{\Phi^-}^Q}^{DQN}(s_{i,t+1}, r_{i,t+1}; \Phi^-) - Q_{\pi_\Phi^Q}^{DQN}(s_{i,t}, a_{i,t}; \Phi) \right] ,$$

$$\mathcal{Q}_{\pi_{\Phi^-}^Q}^{DQN}(s_{i,t+1}, r_{i,t+1}; \Phi^-) = r_{i,t+1} + \gamma Q_{\pi_{\Phi^-}^Q}^{DQN}(s_{i,t+1}, \kappa_{i,t+1}; \Phi^-) , \qquad (16)$$

$$\kappa_{i,t+1} = \arg \max_{\kappa_{i,t+1} \in \psi_{i,t+1}} Q_{\pi_\Phi^Q}^{DQN}(s_{i,t+1}, \kappa_{i,t+1}; \Phi) ,$$

where $\mathcal{Q}_{\pi_{\Phi^-}^Q}^{DQN}$ is the learning target function, $\gamma$ is the discount factor, $\psi_{i,t+1}$ is the possible action space for worker $i$ at time $t+1$, and $\Phi^-$ represents the parameters of the target network, which are updated at a slower pace compared to the policy network to provide a stable target for training. After each training iteration, the target network is updated in a soft manner: $\Phi^- := \tau\Phi + (1-\tau)\Phi^-$, where $\tau$ is the update rate.

The detailed process is illustrated in Algorithm 1, where $\mathbf{1}_j$ represents the vector that only the $j^{th}$ position is 1 and other positions are 0.

---

**Algorithm 1** IDDQN Training Process

---

**Require:** Number of training episodes $E$, number of training steps $T$, mini-batch size $m$, target update rate $\tau$, exploration noise $\epsilon$, final exploration $\epsilon_f$, exploration decay $\delta$, discount factor $\gamma$, model parameters $\Phi$
1: Initialize target networks $\Phi^- \leftarrow \Phi$
2: Initialize replay buffer $\mathcal{B}$
3: **for** $k = 1$ to $E$ **do**
4:     **for** $t = 1$ to $T$ **do**
5:         Calculate Q-value matrix $Y_t$: $y_{i,j,t} = Q_{\pi_\Phi^Q}^{DQN}(s_{i,t}, \mathbf{1}_j; \Phi)$
6:         Select action with exploration noise: $A_t = \text{ILP}(Y_t, \epsilon)$
7:         Observe reward $r_{i,t+1}$ and new state $s_{i,t+1}$ for each worker $i$
8:         Store transition $(s_{i,t}, a_{i,t}, r_{i,t+1}, s_{i,t+1})$ in $\mathcal{B}$
9:         Sample mini-batch of $m$ transitions $(s, a, r, s')$ from $\mathcal{B}$
10:        Compute target Q-value:
11:        $y \leftarrow r + \gamma Q_{\pi_{\Phi^-}^Q}^{DQN}(s_{i,t+1}, \arg\max_{\kappa_{i,t+1} \in \psi_{i,t+1}} Q_{\pi_\Phi^Q}^{DQN}(s_{i,t+1}, \kappa_{i,t+1}; \Phi); \Phi^-)$
12:        Update Q-Network: $\Phi \leftarrow \arg\min_\Phi \frac{1}{m} \sum (y - Q_{\pi_\Phi^Q}^{DQN}(s, a; \Phi))^2$
13:        Update target networks: $\Phi^- \leftarrow \tau\Phi + (1-\tau)\Phi^-$
14:     **end for**
15:     Decay exploration: $\epsilon \leftarrow \max(\epsilon_f, \epsilon\delta)$
16: **end for**

---

### D.2 STAGE 2: TD3 ALGORITHM

The process of our Stage 2 - TD3 training is illustrated in Algorithm 2. In experiment, we follow the vanilla TD3 approach of updating the actor once after updating the critic twice.

---

**Algorithm 2** TD3 Training Process

---

**Require:** Number of training episodes $E$, number of training steps $T$, mini-batch size $m$, policy delay $d$, target update rate $\tau$, exploration noise $\epsilon$, final exploration $\epsilon_f$, exploration decay $\delta$, target policy smoothing noise $\xi$, discount factor $\gamma$, model parameters $\Theta$
1: Initialize target networks $\Theta^- \leftarrow \Theta$
2: Initialize replay buffer $\mathcal{B}$
3: **for** $k = 1$ to $E$ **do**
4:     **for** $t = 1$ to $T$ **do**
5:         Select action with exploration noise: $A_t = \text{Actor}(S_t; \Theta, \epsilon)$
6:         Observe reward $R_{t+1}$ and new state $S_{t+1}$
7:         Store transition $(S_t, A_t, R_{t+1}, S_{t+1})$ in $\mathcal{B}$
8:         Sample mini-batch of $N$ transitions $(S, A, R, S')$ from $\mathcal{B}$
9:         Compute target action with smoothing noise: $A' \leftarrow \text{Actor}(S; \Theta^-, \xi)$
10:       Compute target Q-value: $y \leftarrow r + \gamma \min_{i=1,2} Q^{TD3}_{\pi^T_{\Theta^-}, i}(S', A'; \Theta^-)$
11:       Update critics: $\Theta \leftarrow \arg\min_\Theta \frac{1}{m} \sum [(y - Q^{TD3}_{\pi^T_\Theta, 1}(S, A; \Theta))^2 + (y - Q^{TD3}_{\pi^T_\Theta, 2}(S, A; \Theta))^2]$
12:       **if** $t \mod d == 0$ **then**
13:         Update actor using deterministic policy gradient:
14:         $\nabla J(\Theta) = \frac{1}{m} \sum (Q^{TD3}_{\pi^T_\Theta, 1}(S, A; \Theta) - B)\nabla_\Theta \log \pi^T_\Theta(A_t|S_t), \quad (A = \text{Actor}(S; \Theta))$
15:         Update target networks: $\Theta^- \leftarrow \tau\Theta + (1-\tau)\Theta^-$
16:       **end if**
17:     **end for**
18:     Decay exploration: $\epsilon \leftarrow \max(\epsilon_f, \epsilon\delta)$
19: **end for**

---

## E  EXPERIMENT DETAILS

### E.1  EXPERIMENT CONFIGURATIONS

Our model was trained using the PyTorch framework [36] on a workstation running Windows 11, equipped with an Intel(R) Core(TM) i7-14700KF processor and an NVIDIA RTX 4080 graphics card. The detailed model configurations are shown as Table 3. During the training phase, the model utilized approximately 8.03 GB of GPU memory. For optimization, we employed the Adam optimizer with an initial learning rate of $10^{-4}$ and a decay rate of 0.99. In Stage 1, the batch size was set to 256, while in Stage 2, it was reduced to 16, due to a sharp decrease in sample amount. Additionally, optimization was performed once every 4 time steps, and in Stage 2, the actor was updated once for every two updates of the critic.

Table 3: Model Configurations

| Configuration | Our Setting |
|---|---|
| Hidden Dimension | 64 (Actor) / 128 (Critic) |
| Attention Heads | 4 |
| BERT Layers | 3 for Each |
| Dropout Rate | 0.1 |
| Optimizer | Adam |
| Learning Rate | $10^{-4}$ |
| Scheduler | ExponentialLR |
| Learning Rate Decay | 0.99 |
| Batch Size | 256 (Stage 1) / 16 (Stage 2) |
| Exploration Rate | $0.99 \rightarrow 0.0005$ |
| Updating Rate of Target Network | 0.005 |
| Discount Factor | 0.99 |

### E.2 Simulation Setup

In the simulation, we follow previous works [23; 9; 1], setting the total number of drivers to 1,000, with each car having a capacity of 3 passengers. The maximum waiting time for customers is set to 5 minutes.Each episode lasts 30 minutes, divided into 30 time steps, where each step determines the operations for the subsequent minute. For the TSP route optimization and time estimation, we utilize the OSRM simulator [30], with a default traveling speed of 60 km/h. For solving the bipartite matching, we use the Hungarian algorithm [34], provided in SciPy [52].

We train the model using data from 19:00 to 19:30 on July 17, 2024, which includes 3,726 valid orders, and we test the trained model during other time periods on July 17, 2024, including 14:00-14:30 (2,850 valid orders), 17:00-17:30 (3,577 valid orders), 20:00-20:30 (3,114 valid orders), 21:00-21:30 (4,264 valid orders), and 22:00-22:30 (4,910 valid orders), where the order amount range from 2,850 to 4,264.

The evaluation metrics include:

- **Served Rate**: the rate of confirmed trips relative to the total trip requests initiated by customers.
- **Delivery Time**: the time taken to serve a trip from origin to destination.
- **Detour Time**: the extra time spent on delivery beyond the minimum delivery time (i.e., the time if the vehicle only serves this trip without bundling other trips).
- **Pickup Time**: the waiting time for customers between the trip confirmation and the arrival of the vehicle at the trip origin.
- **Confirmation Time**: the waiting time for customers from when they initiate the trip request to when the platform assigns the trip to a vehicle.

### E.3 Introduction of Comparative Methods

The methods using in our comparative experiment can be mainly divided into three categories:

- **Independent MARL:** The DeepPool [1] and BMG-Q [23] utilize a similar IDDQN method as described in Section 3.3.1, with BMG-Q employing GAT [51] to capture the relationships among neighboring agents. Additionally,in the original paper fo DeepPool, the authors used CNN. However, due to differences in the observation space of our task, we replaced it with MLP.
- **Centralized Training Decentralized Execution (CTDE):** The HIVES [38] framework introduces a QMIX [38] based method to address the shortcomings of IDDQN, specifically the inadequacy of treating the global Q-value as a simple summation of the individual Q-values of each agent. Enders et al. [9] propose a MASAC [14] based approach, allowing each driver to choose whether to accept an order, thereby preventing low-profit orders from negatively impacting the global income.
- **Centralized Training and Centralized Execution (CTCE):** CEVD [3], based on VD [45], innovatively combines the Q-values of each agent with those of their neighbors to create a new type of Q-value, akin to the motivation behind BMG-Q.

Overall, most of these methods attempt various strategies to enhance each agent's awareness of the global state, facilitating better cooperation. In contrast, our method directly transforms the formulation into a centralized single-agent reinforcement learning approach.

It is noteworthy that these Independent and CTDE MARL dispatching methods differ slightly from general MARL methods. In order dispatching, one order cannot be assigned to multiple workers, making it necessary to employ some centralized mechanism to achieve this. We refer to them as independent MARL and CTDE methods because they can directly calculate their own Q-values or action probabilities using their own or neighboring states. Conversely, CEVD must calculate the primary Q-value of each agent separately and then combine those primary Q-values with their neighbors to obtain a final Q-value for each agent.

Through the experimental results in Fig. 3, we observe that DeepPool [1], serving as one of the earliest benchmarks, demonstrates relatively stable and good performance, suggesting the simplicity and effectiveness of IDDQN features. In contrast, BMG-Q [23] significantly improves performance by utilizing FAT to capture neighboring information. As for HIVES [38] and CEVD [3], while they exhibit relatively good performance in the early stages of training—likely due to their hierarchical structure and centralized training methods—their performance becomes unstable in later stages, with rewards even starting to decrease. This instability may stem from the hierarchical approach not

adequately addressing the large network input of the mixture network in QMIX and the lazy agent problem in VD. Additionally, their centralized training approach faces the same data scarcity issues as our method, making convergence more challenging. For Enders et al. [9], we note that their method shows worse performance than others. This may be related to their state processing method during training, where they replace the next state in the replay buffer with the request state from the current state to maintain a consistent agent count across two successive time steps, which appears to be a strong assumption. Finally, for the last three methods, their original papers primarily focus their reward functions on the serving order amount, without incorporating additional terms like ours (which also considers income, outcome, and user satisfaction levels). This makes our scenario more complex and may further reduce the performance of their methods in our setting.

### E.4 ADDITIONAL EXPERIMENT RESULT

The detailed experimental results across different time periods are shown in Fig. 6, while the weighted average numerical results are presented in Table 4. For each model in each scenario, we repeat the experiment three times, and the error bars in the figure represent the standard deviation. We observe that our Triple-BERT achieves the highest reward across all scenarios, with the advantage becoming more pronounced as the order volume increases. Triple-BERT primarily optimizes the service rate and pickup time, significantly outperforming other methods.

For delivery time and detour time, the figures only account for completed orders, as the status of unfinished orders is uncertain, which may introduce some bias in the detailed values. In terms of these two metrics, Triple-BERT clearly performs better in high order volume scenarios, but not in low order volume scenarios. This may be due to the relatively low conflict caused by MARL in low order scenarios, while in high order scenarios, both the observation and action spaces increase sharply, making it challenging for MARL to find optimal solutions.

Lastly, we note that our method and the approach by Enders et al. [9] exhibit higher confirmation times. This may be attributed to both methods having an explicit rejection action (i.e., choosing no order), unlike the other methods. While this mechanism can lead to higher confirmation times, it also enables the model to discard negative profit orders and reserve some orders for currently unavailable workers.

Table 4: Average Performance under Multiple Periods: **Bold** entries represent the best results.

| Method | Reward | Service Rate | Delivery Time | Detour Time | Pickup Time | Confirmation Time |
|---|---|---|---|---|---|---|
| **DeepPool** [1] | 12723.85 | 0.91 | 11.53 | 2.47 | 7.77 | 0.06 |
| **BMG-Q** [23] | 13036.29 | 0.92 | **10.57** | **1.90** | 7.61 | 0.10 |
| **HIVES** [16] | 12365.11 | 0.89 | 11.04 | 2.28 | 7.99 | **0.03** |
| **Enders et al.** [9] | 12041.62 | 0.90 | 12.28 | 2.90 | 7.94 | 0.80 |
| **CEVD** [3] | 13157.96 | 0.94 | 11.36 | 2.31 | 7.37 | 0.06 |
| **Triple-BERT** | **14730.48** | **0.98** | 11.53 | 2.52 | **5.73** | 0.13 |
| **w/o stage 1** | 10665.02 | 0.87 | 11.92 | 2.72 | 9.36 | 0.68 |
| **w/o normalization** | 10839.33 | 0.88 | 12.50 | 2.85 | 9.10 | 0.24 |

### E.5 EXPANDED GENERALIZATION AND EXTENSIBILITY EXPERIMENT

In this section, we first further prove the generalization capacity of our method in more testing scenarios, using the scenarios of other days, not just the other periods in the same day as the last section. The result is shown in Table 5. Specifically, we compared our Triple-BERT with other methods using the testing scenario from July 16 to July 18, 2024, at 6 PM. The results indicate that our Triple-BERT achieved the highest reward among all scenarios, highlighting its strong generalization capacity.

Then, we aim to prove the extensibility and scalability of our method. As a result, we compared its performance against other approaches as the driver count increased to 1,500 and 2,000 during a high concurrency period. In this scenario, the order volume reached 6,775, which we synthesized by combining orders from two different periods. The result is shown in Table 6. The results indicate that our Triple-BERT model achieves the highest reward across various scenarios, without the need for retraining.

Table 5: Reward of Different Methods Under Different Days

| Scenario | DeepPool [1] | BMG-Q [23] | HIVES [16] | Enders et al. [9] | CEVD [3] | Triple-BERT |
|---|---|---|---|---|---|---|
| **7.16 (4,451 orders)** | 13,473 | 14,121 | 12,070 | 12,142 | 14,226 | **16,831** |
| **7.17 (4,125 orders)** | 13,204 | 13,424 | 12,232 | 12,208 | 14,145 | **16,145** |
| **7.18 (3,635 orders)** | 12,679 | 13,067 | 12,397 | 12,268 | 13,336 | **14,819** |

Table 6: Reward of Different Methods During High Concurrency Period among Different Driver Amounts

| Driver Amount | DeepPool [1] | BMG-Q [23] | HIVES [16] | Enders et al. [9] | CEVD [3] | Triple-BERT |
|---|---|---|---|---|---|---|
| **1,500** | 21,090 | 22,333 | 19,587 | 19,546 | 23,092 | **27,458** |
| **2,000** | 25,316 | 25,650 | 25,713 | 25,394 | 26,207 | **28,273** |

Additionally, to prove the practical application potential of our method, we test the computation time with driver amounts ranging from 1,000 to 2,000 and order amounts from 300 to 500 at a single time step, shown in Table 7. (In our real-world dataset, we typically observe that the order amount does not exceed 200 at any single step.) The results indicate that the decision time remains consistently under 0.2 seconds across all scenarios. Additionally, the order and driver counts have minimal impact on computation time. This suggests that, for the current simulation, most of the processing time is allocated to the simulator's operations rather than to decision computation cost.

Table 7: Decision Time of Different Driver and Order Amounts (unit: seconds)

| Driver Amount \ Order Amount | 300 | 400 | 500 |
|---|---|---|---|
| **1,000** | 0.1801 | 0.1839 | 0.1870 |
| **1,500** | 0.1806 | 0.1820 | 0.1830 |
| **2,000** | 0.1789 | 0.1829 | 0.1809 |

To further demonstrate the generalizability of our method, we conducted an expanded experiment using High Volume For-Hire Vehicle (FHV) trip data from Queens, New York City [47]. We chose not to continue with the yellow taxi data, as its primary operational area is Manhattan. Unlike the capital-intensive region of Manhattan, Queens has a significantly lower trip volume, despite its area being about five times larger. Consequently, the data distribution in Queens presents a markedly different challenge: while the number of orders decreases, the distances between them tend to increase, leading to greater difficulties for ride-hailing services. To adapt to this scenario, we set the driver count to 500 while keeping all other settings consistent.

The detailed experimental results are presented in Table 8, using data from 19:00 to 19:30 on July 17, 2024, which includes 2,024 valid orders. We observe that our Triple-BERT maintains its SOTA performance, primarily optimizing the assignment and improving rewards by serving more orders and reducing pickup times. However, this inevitably leads to a slight increase in delivery and detour times due to the bundling of more orders.

Additionally, compared to the results in Manhattan, as shown in Table 4, we noted that the average pickup time in Queens is significantly longer. While the decrease in the number of drivers may contribute to this, it is not the main factor, as the order volume has also declined significantly. The primary reason lies in the larger area and more dispersed order distribution in Queens. This leads to a substantial penalty in the pickup time term of the reward function, resulting in minimal differences in the rewards among different policies. Such challenging scenarios further hinder the efficient exploration and learning of centralized MARL methods like HIVES and CEVD.

### E.6 EXPANDED ABLATION STUDY

In this section, we aim to further illustrate the efficacy of removing the positional embedding in BERT and the utilization of ARL style attention-based encoder for better feature extraction.

Table 8: Performance of Different Methods in Queens, New York City [47]

| Method | Reward | Service Rate | Delivery Time | Detour Time | Pickup Time | Confirmation Time |
|--------|--------|--------------|---------------|-------------|-------------|-------------------|
| **DeepPool** [1] | 5222.85 | 0.64 | 11.24 | 1.84 | 12.30 | **0.21** |
| **BMG-Q** [23] | 5362.00 | 0.66 | 9.63 | 1.08 | 12.98 | 0.27 |
| **HIVES** [16] | 3560.80 | 0.60 | **8.30** | **0.36** | 14.67 | 0.41 |
| **Enders et al.** [9] | 4543.68 | 0.61 | 10.41 | 0.85 | 13.39 | 2.25 |
| **CEVD** [3] | 4388.83 | 0.62 | 11.61 | 1.33 | 13.74 | 0.29 |
| **Triple-BERT** | **5577.83** | **0.72** | 9.07 | 0.90 | **11.32** | 0.23 |

For the positional embedding, the rationale behind eliminating the positional embedding includes:

- **Incorporated Position Information in State Space**: The coordinates of each vehicle and order are included as part of the state input, making additional positional embeddings unnecessary.
- **Nature of Vehicle and Order Dispatching**: In ride-sharing tasks, the optimal assignment should be independent of the input sequence order. By removing the positional embedding, we ensure that all positions are homogeneous, thus realizing this property.
- **Scalability Considerations**: Utilizing positional embeddings in BERT requires a predefined maximum input length before model training, which cannot be altered later. In scenarios of extreme concurrency, the number of orders may exceed this maximum length, potentially compromising model efficacy.

To further prove it, we compare the model performance with and without the positional embedding as shown in Table 9. The results indicate that positional embedding introduces additional interference, resulting in lower performance. This is particularly evident in generalization problems: when using positional embeddings, Triple-BERT only outperforms previous MARL SOTA in training scenarios, but not in testing scenarios. Additionally, when the order amount exceeds the training scenario, the maximum length restriction hinders the model's effectiveness.

Table 9: Reward of Tripe-BERT w/ and w/o Positional Embedding (PE)

| PE \ Order Amount | 3,726 (train) | 2,850 | 3,114 | 3,577 | 3,910 | 4,264 |
|-------------------|---------------|-------|-------|-------|-------|-------|
| **w/** | 14,092 | 10,679 | 11,431 | 12,841 | × | × |
| **w/o** | **15,388** | **11,148** | **13,483** | **14,477** | **16,335** | **17,366** |

To illustrate the efficiency of the encoder, we compare our methods and others when using the attention-based encoder and vanilla MLP-based encoder. The result is shown in Table 10. The results indicate that the ARL-based encoder significantly enhances the performance of our Triple-BERT alongside independent MARL methods such as DeepPool and BMG-Q. However, this improvement does not extend to CTDE and centralized MARL approaches like HIVES, Enders et al., and CEVD. The underlying reason is that CTDE and centralized MARL methods in ride-sharing primarily suffer from the CoD in the critic network, an issue that cannot be mitigated by a more powerful feature extractor. We believe that the improvements observed in DeepPool, BMG-Q, and our Triple-BERT effectively enhance the efficiency of our designed encoder. Furthermore, even with the ARL-based encoder, our Triple-BERT consistently outperforms all previous methods, underscoring the superiority of our approach. Additionally, even with the simple MLP-based encoder, Triple-BERT still outperforms previous MARL methods, illustrating the robustness of our centralized SARL method.

Table 10: Reward of Different Methods Under Different Encoder

| Encoder | DeepPool [1] | BMG-Q [23] | HIVES [16] | Enders et al. [9] | CEVD [3] | Triple-BERT |
|---------|--------------|------------|------------|-------------------|----------|-------------|
| **Vanilla MLP** | 13,332 | 13,539 | **13,126** | 12,670 | **13,746** | 14,967 |
| **ARL-based** | **14,570** | **13,879** | 13,095 | **12,706** | 13,724 | **15,388** |

## F RELATED WORK

### F.1 ORDER DISPATCH IN RIDE-SHARING TASK

Order dispatch methods in ride-sharing can be primarily categorized into model-based and RL-based approaches. Model-based methods often rely on early assumptions that all order information is known in advance [59] or neglect potential future dynamics, resulting in impractical or myopic solutions. Later research focused on modeling and capturing environmental uncertainties for practical applications [2]. However, accurately characterizing these complexities in the ride-sharing market remains a challenging task.

In contrast, model-free RL methods free researchers from these constraints, allowing agents to interact with and learn from the environment independently. Given the vast action and observation spaces, most studies adopt a MARL paradigm (some referring to their methods as decentralized RL) to effectively manage these challenges [37]. Similar to standard MARL, the methods adapted for ride-sharing can be divided into Decentralized Execution (DTDE, also known as independent MARL), Centralized Training with Decentralized Execution (CTDE), and Centralized Training with Centralized Execution (CTCE) [24]. DTDE methods, while widely used in early studies [1; 10], suffer from low cooperation since each agent perceives others merely as part of the environment, often leading to instability during training. Hu et al. [23] introduced the GAT [51] to enable each agent to consider the information of its neighbors, thereby reducing overestimation issues and improving system performance. Other studies have shifted towards CTDE and CTCE paradigms to promote effective cooperation. For instance, Enders et al. [9] proposed a delayed matching method based on MASAC [14], allowing agents to decide whether to accept assigned orders at each step. Bose et al. [3] developed a VD-based method [45] that utilizes a global reward to foster cooperation, although it faces the challenge of lazy agents. Furthermore, Hao et al. [16] applied QMIX [38], while Hoppe et al. [19] employed COMA [11] alongside a mix of global and local rewards to address existing issues. However, many of these centralized methods require a centralized critic, reintroducing the challenge of CoD. To address this, Li et al. [26] proposed a Mean Field MARL framework [60], and Zhao et al. [66] developed the GRPO [39] and OSPO methods based on the homogeneous properties among agents, although these assumptions are not always met in practice.

Additionally, several studies began to explore more practical scenarios that integrate order dispatch with other tasks, such as repositioning, price setting, and multi-modal transportation[5]. For example, Zhang et al. [63] considered the joint optimization of order dispatch and price setting, which can significantly influence customer demand. Similarly, Haliem et al. [15] and Ge et al. [13] examined repositioning strategies based on these factors. Hu et al. [21; 22] analyzed order assignment in joint delivery scenarios involving cars, subways, and Unmanned Aerial Vehicles (UAVs), while other works, such as Singh et al. [41], have studied multi-hop transportation. Furthermore, some studies have examined fairness considerations. For instance, Zhang et al. [62] introduced mutual information in the reward function to address challenges related to unusual order distributions and improve platform income. Zhao et al. [65; 67] investigated algorithmic discrimination by considering joint order assignments and payment settings for workers.

*Overall, current MARL-based ride-sharing methods face a fundamental dilemma: centralized training paradigms foster cooperation at the cost of exacerbating the CoD, whereas decentralized approaches mitigate CoD but suffer from non-stationarity and poor coordination, ultimately limiting system performance. To navigate this dilemma, our paper offers a novel perspective by directly utilizing a centralized SARL framework to leverage complete global information, where we then develop efficient techniques, such as action decomposition, to directly address the ensuing CoD challenge.*

### F.2 COOPERATIVE MULTI-AGENT REINFORCEMENT LEARNING

Cooperative MARL has a wide range of applications, including power control, robot fleet management, and ride-sharing tasks [61]. According to a recent survey [35], these methods can be categorized into five types: independent learning, centralized critic, value decomposition, consensus, and communication.

---

[5]To avoid confusion, we would like to clarify that the concept of multi-modal in transportation differs from that in computer science. In transportation, "multi-modal" refers to a travel model that utilizes multiple vehicles, such as taxis, buses, and subways, regardless of the model input.

Early-stage works primarily focused on independent learners, which represent the simplest realization of MARL. By viewing each agent as an independent entity and others as part of the environment, standard SARL methods can be easily transferred to MARL scenarios. However, this approach can lead to unstable environments, as the policies of other agents continuously change. Additionally, it often results in local optima, as each agent aims to maximize its own reward, neglecting the cooperation necessary to optimize global reward. Despite efforts such as Hysteretic Q-Learning [32], challenges remain in large-scale environments and sparse reward situations.

To overcome these challenges, the CTDE and CTCE paradigms have been widely explored in centralized critic and value decomposition MARL methods. In the case of centralized critics, early research focused on adapting actor-critic methods like PPO, SAC, and DDPG by replacing the critic with a centralized one. During training, the critic receives input from all agents, thus addressing the instability problem, although it is not needed during execution, allowing actors to function independently. To further enhance this paradigm, attention-based critics were introduced to capture the relationships among agents [31], inspired by the Transformer architecture [50]. Foerster et al. [11] proposed COMA to resolve the credit assignment problem by introducing a counterfactual baseline, which underwent further improvements in subsequent research [25]. Additionally, value decomposition methods aim to effectively distribute global rewards among agents, enabling them to optimize the overall reward rather than just their individual rewards. Starting from VD [45], which struggles with lazy agents due to the simple addition of Q-values, many efforts have been made to enhance the representation capacity of functions that combine global and individual Q-values. Notable examples include QMIX [38], VDAC [43], and QTRAN [42]. *However, most of these methods encounter the challenge of CoD when the number of agents increases, particularly in scenarios like our ride-sharing task, where the number of agents can exceed hundreds or even thousands.*

Consensus and communication methods, developed later, strive to find a balance between low cooperation and the CoD challenge. Here, agents only exchange information with their neighbors or selected agents. Consensus-based methods utilize sparse communication to achieve policy consensus among agents, often with convergence guarantees under linear approximators [49]. However, many of these methods often rely on multiple rounds of communication, leading to practical challenges in the ride-sharing task, which has high demands for real-time performance. In contrast, communication-based methods focus on designing efficient mechanisms for determining what information to send and to whom. For instance, Sukhbaatar et al. [44] proposed CommNet, which broadcasts each agent's hidden features derived from their observations. However, similar to Mean Field approaches [60] in MARL, this method considers only the average influence of others, neglecting the specific relationships among agents. Subsequent attention-based methods were introduced to evaluate the significance of information from different agents [7; 27]. However, the training difficulties associated with communication-based methods are relatively high, particularly as early-stage communication often yields little meaningful information. Furthermore, many communication methods face a trade-off between cooperation performance, communication costs, and the content of communication—issues closely related to CoD. Even recent methods focusing on efficient communication mechanism design [55] still encounter challenges in our ride-sharing scenario with extremely high agent volumes. For example, [28] proposed a novel communication protocol based on exponential graphs, guaranteeing global message exchange within $\lceil \log_2(N - 1) \rceil$ steps. However, this number can reach 10 in our experiment with 1,000 agents, which is relatively high compared to the total time step horizon of 30. Additionally, communication-based methods represent an ongoing challenge, as determining what content to communicate and to whom poses risks associated with credit assignment. *Despite these advancements, most consensus and communication methods still encounter challenges such as credit assignment and a lack of convergence theory. In contrast, the SARL-based method circumvents these problems by directly capturing global information. Given that ride-sharing is fundamentally a centralized decision task, we consider the SARL-based method to be a promising solution that overcomes the limitations of previous MARL-based approaches.*

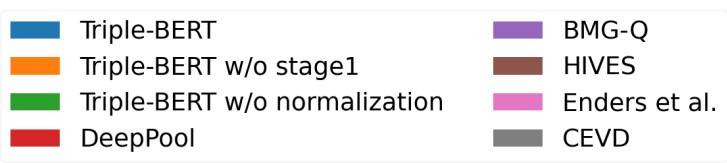

(a) Legend

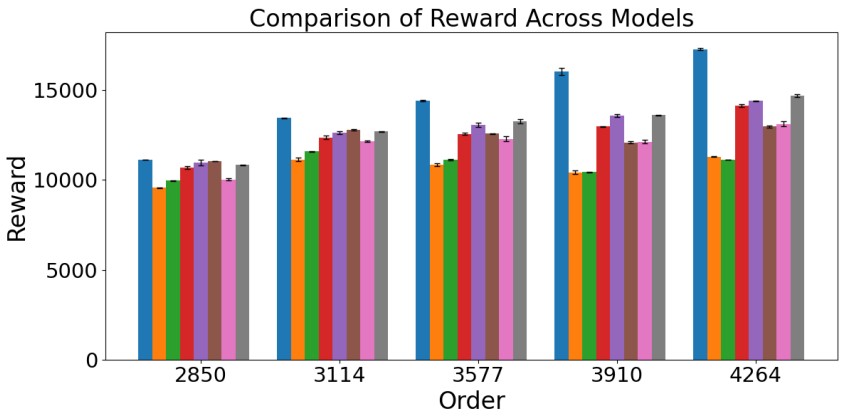

(b) Accumulative Reward

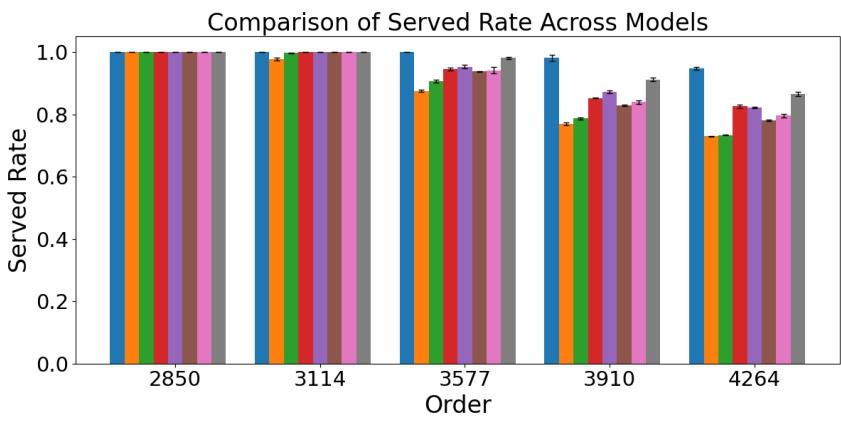

(c) Service Rate

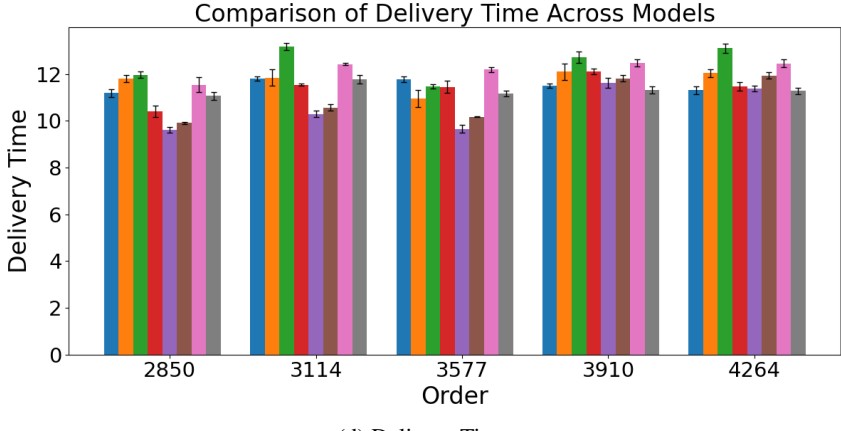

(d) Delivery Time

Figure 6: Detailed Evaluation Results

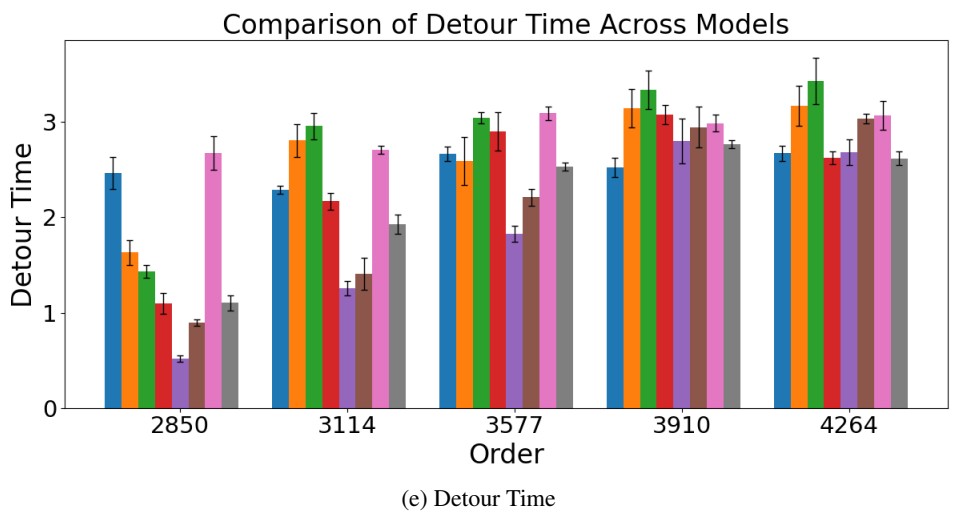

(e) Detour Time

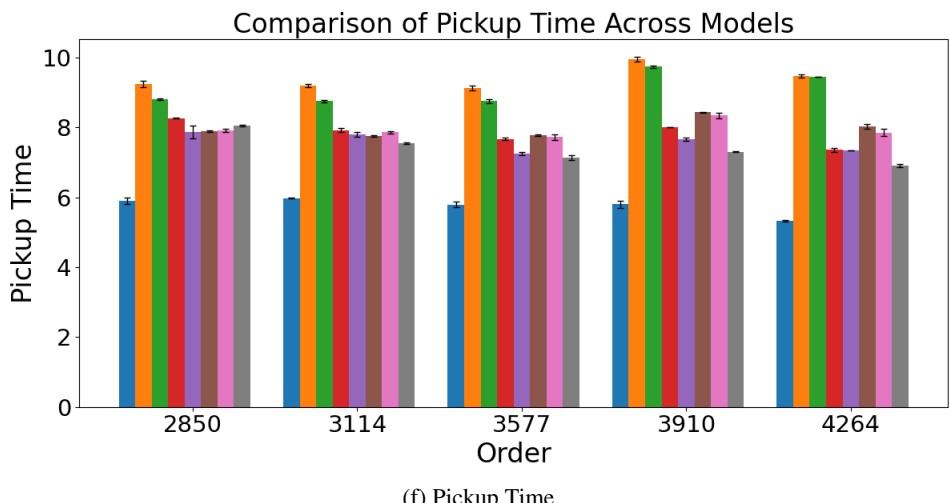

(f) Pickup Time

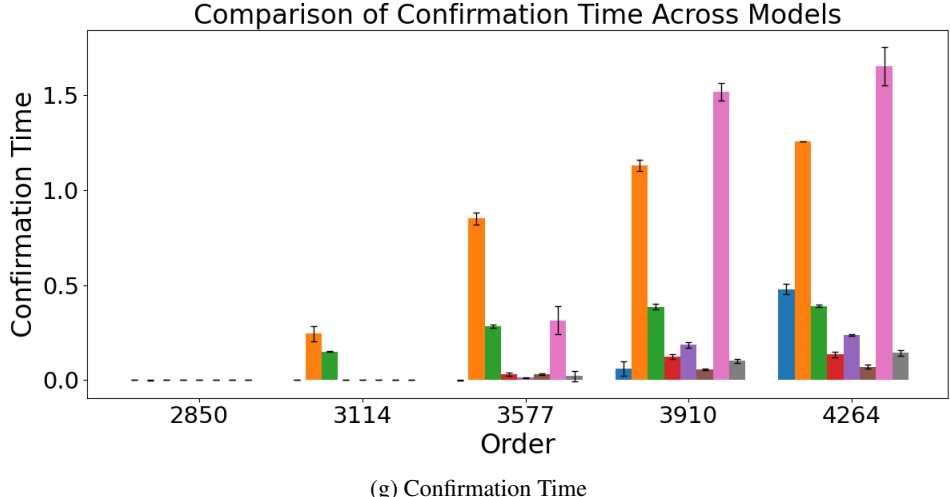

(g) Confirmation Time

Figure 6: Detailed Evaluation Results

