# OpenReview forum: "Triple-BERT: Do We Really Need MARL for Order Dispatch on Ride-Sharing Platforms?"
_ICLR.cc/2026/Conference — ICLR 2026 Oral_

### Official Review · Reviewer_PcM6 · 2025-10-29

**Soundness:** 3
**Presentation:** 3
**Contribution:** 2
**Rating:** 8
**Confidence:** 3

**Summary:**

This paper proposes Triple-BERT, a centralized single-agent reinforcement learning approach tailored for the order dispatch problem on ride-sharing platforms, aiming to address the challenges faced by traditional multi-agent reinforcement learning (MARL) methods in handling large-scale order dispatch scenarios. The paper introduces an innovative network architecture and training strategy that significantly enhances the efficiency of order dispatch and platform performance. Overall, the research is innovative, and the experimental results are impressive, offering substantial theoretical and practical contributions to the ride-sharing domain.

**Strengths:**

1. Clear Problem Definition
2. Triple-BERT innovatively addresses the large action and observation spaces through an action decomposition strategy and BERT network  The incorporation of the QK-Attention mechanism significantly reduces computational complexity.

**Weaknesses:**

While the experimental results are significant, the theoretical analysis in the paper is relatively weak. For instance, a deeper exploration of why the action decomposition strategy works effectively and its theoretical foundations could be further elaborated.
In the paper, the literature review on related work appears to be somewhat limited. For instance, significant contributions such as "Future Aware Pricing and Matching for Sustainable On-Demand Ride Pooling" , and studies exploring "MARL-Based Pricing Strategy via Mutual Attention for Mobility-on-Demand (MoD) Systems with Ridesharing and Repositioning," as well as "Mutual Information as Intrinsic Reward of Reinforcement Learning Agents for On-Demand Ride Pooling,"

**Questions:**

No

---

> ### Author Response · Authors · 2025-11-17
> **Response to Reviewer PcM6**
>
> We are grateful for your thoughtful feedback. Your assessment is valuable for us to polish our paper. We are glad to reply to your questions.
>
> > W1: While the experimental results are significant, the theoretical analysis in the paper is relatively weak. For instance, a deeper exploration of why the action decomposition strategy works effectively and its theoretical foundations could be further elaborated.
>
> Thank you for your insightful comment. We have revised the theoretical analysis of the action decomposition strategy in Section 3.3.2 (i) Actor part, and in Appendix C.
>
> In standard discrete Actor-Critic (AC) methods, the actor network directly generates the action probability vector and then samples actions from it. However, in the ride-sharing task, due to the high volume of drivers and orders, the action space can reach nearly $10^{30}$ (as demonstrated in Appendix A), making it impossible to generate and learn such an actor probability vector. To tackle this challenge, we propose an action decomposition method, expressed as (Eq. 5):
>
> $$
> \pi_{\Theta}^T(A_t|S_t) = \text{z} \left(\prod_{i,j \in \text{h}(A_t)} \mathscr{P}_{i,j,t} \right)
> $$
>
> where $\mathscr{P_{i,j,t}}$ represents a virtual probability of assigning order $j$ to courier $i$ at time $t$, and $\text{z}(\cdot)$ is an increasing function that maps the virtual probability to the real action probability of the policy $\pi_{\Theta}^T$.
>
> For optimization, we derive the policy in Appendix C (Eq. 14):
>
> $$
> \nabla_\Theta \text{J}(\Theta) = \mathbb{E}_{\pi_{\Theta}^T} \left[ \left(\text{Q}^{TD3}_{\pi_{\Theta}^T}(S_t,A_t) - B\right) \mathcal{E}_{\text{z}(x),x}|_{x=\prod_{i,j \in \text{h}(A_t)} \mathscr{P}_{i,j,t}} \nabla_\Theta \sum_{i,j \in \text{h}(A_t)} \log \mathscr{P}_{i,j,t} \right]
> $$
>
> To simplify optimization, we define the format of $\text{z}(\cdot)$ as $z(x) = ax^b$, where $a, b > 0$. This formulation has the advantage that the elasticity of $\text{z}(\cdot)$ is a positive constant, i.e., $\mathcal{E}_{\text{z}(x),x} = ab$. Thus, we can state, corresponding to Eq. 7 in Section 3.3.2.:
>
> $$
> \nabla_\Theta \text{J}(\Theta) \propto \mathbb{E}_{\pi_{\Theta}^T} \left[ \left(\text{Q}^{TD3}_{\pi_{\Theta}^T}(S_t,A_t) - B\right) \nabla_\Theta \sum_{i,j \in \text{h}(A_t)} \log \mathscr{P}_{i,j,t} \right]
> $$
>
> By defining the virtual action probability $\mathscr{P}$ and mapping function $\text{z}(\cdot)$, we connect the output of the network to the policy $\pi_{\Theta}^T$. Essentially, we restrict the policy space to a smaller class as defined by Eq. 5 to facilitate optimization and application. While this method may impose restrictions on the representational capacity of the actor, the benefits derived from our centralized approach outweigh these limitations, as evidenced by the state-of-the-art performance of the proposed Triple-BERT.
>
> We have revised the corresponding sections in the manuscript (Lines 363-390 on Pages 7-8, Lines 894-904 on Page 17) in accordance with your comment. Thank you again for helping us refine our work.
>
> > W2:  In the paper, the literature review on related work appears to be somewhat limited. For instance, significant contributions such as "Future Aware Pricing and Matching for Sustainable On-Demand Ride Pooling" , and studies exploring "MARL-Based Pricing Strategy via Mutual Attention for Mobility-on-Demand (MoD) Systems with Ridesharing and Repositioning," as well as "Mutual Information as Intrinsic Reward of Reinforcement Learning Agents for On-Demand Ride Pooling,"
>
> Thank you for your insightful suggestion. We have revised our Related Work section (Appendix F) to include various types of MARL-based ride-sharing methods and cooperative MARL methods. For order dispatch methods in ride-sharing tasks, we first review different types, including CTCE, CTDE, and DTDE. Furthermore, we briefly introduce more practical and comprehensive works in ride-sharing, considering joint order assignment alongside other tasks such as repositioning, price setting, multi-modal transportation, and fairness concepts. We appreciate your suggestion of the three valuable works, and we have included them in our Related Work section. (Page 25-26)
>
> ---
>
> Finally, we express our heartfelt gratitude for your insightful questions and suggestions. They have been invaluable to our work. We hope our efforts will adequately address your concerns.

---

### Official Review · Reviewer_TZpd · 2025-10-29

**Soundness:** 3
**Presentation:** 3
**Contribution:** 2
**Rating:** 6
**Confidence:** 5

**Summary:**

This paper presents a centralized single agent reinforcement learning method for large-scale order dispatching on ride-sharing platforms. In particular, the authors provide a two-stage training strategy to achieve good performance, i.e., 1) the first stage uses IDDQN for MARL pre-training that yields good feature extraction; 2) the second stage employs TD3 for SARL fine-tuning to get the final order dispatch result. In addition, the network backbone design consists of BERT, MLP and QK-attention. Experimental results validated the effectiveness of the proposed method.

**Strengths:**

(1) The motivation is well presented of using a two-stage training strategy for order dispatch on ride-sharing platforms.

(2) The explanations and illustrations are mostly clear and intuitive of the action space, the actor sub-network, the critic sub-network, and the solving algorithms.

**Weaknesses:**

(1) The main contribution is the two-stage training strategy but some other possible approaches are not analyzed. In particular, the title claims MARL is not required but used as pre-training, why not directly consider SARL without MARL pre-training.

(2) On Page 5, Line 236-237, the positional embedding is discarded due to permutation invariance of the input sequence. This is not a typical practice in the literature, it could be better if an ablation study is conducted for the difference. Besides, if permutation invariance is important for the whole framework, using GNN to model the input could be a better choice as done in [16,34].

(3) On Page 5, Line 269, this guarantees a unique solution. No proof or explanation on this point.

(4) On Page 6, Line 323-324, the authors assume that unsatisfactory independent MARL can provide a good starting point. This point is not proved or analyzed to some extent.

(5) On Page 6, Line 299-300, the authors claim that IDDQN is chosen for the stage-1 pre-training. However, in Table 1, the authors present the proposed Triple-BERT uses TD3 as a difference to comparison methods. It is a bit confusing since the whole framework is a two-stage approach and it is inappropriate to only highlight the stage-2 as a contribution.

**Questions:**

No.

---

> ### Author Response · Authors · 2025-11-17
> **Response to Reviewer TZpd (Part 1)**
>
> We are grateful for your thoughtful feedback. Your assessment is valuable for us to polish our paper. We are glad to reply to your questions.
>
> > W1: The main contribution is the two-stage training strategy but some other possible approaches are not analyzed. In particular, the title claims MARL is not required but used as pre-training, why not directly consider SARL without MARL pre-training.
>
> Thank you for your insightful question. As mentioned in the paper, a key challenge when transferring the paradigm from MARL to SARL is data scarcity. In MARL, we can collect $n$ samples at each step, where $n$ is the number of agents. However, in SARL, we only obtain one sample per step, which hinders efficient training. As shown in our experiments (Fig. 3), without the MARL pre-training stage, the proposed method performs poorly. Typically, a reinforcement learning algorithm requires millions of steps to converge, corresponding to over 30,000 episodes in our scenario, which would take more than 20 days of simulation and is not practical. In contrast, by utilizing our MARL pre-training and SARL fine-tuning mechanism, we only need a total of 2,000 episodes, significantly improving training efficiency.
>
> [1] Mnih V, Kavukcuoglu K, Silver D, et al. Human-level control through deep reinforcement learning. *Nature*, 2015, 518(7540): 529-533.
>
> > W2: On Page 5, Line 236-237, the positional embedding is discarded due to permutation invariance of the input sequence. This is not a typical practice in the literature, it could be better if an ablation study is conducted for the difference. Besides, if permutation invariance is important for the whole framework, using GNN to model the input could be a better choice as done in [16,34].
>
> Thank you for your insightful comment. First, regarding the positional embedding issue, we have included a comparison of using and removing it in Appendix E.6 (Table 7). The results indicate that positional embedding does not benefit model training. On the contrary, it introduces additional interference, which results in lower performance, especially in testing set. Additionally, when the order amount exceeds the training scenario, the maximum length restriction from positional embedding hinders the model's effectiveness.
>
> | Order Amount     | w/o positional embedding | w/ positional embedding |
> | :--------------- | :----------------------- | :---------------------- |
> | 3,726 (training) | **15,388** (+8.84%)      | 14,092                  |
> | 2,850            | **11,148** (+4.39%)      | 10,679                  |
> | 3,114            | **13,483**(+17.95%)      | 11,431                  |
> | 3,577            | **14,477**(+12.74%)      | 12,841                  |
> | 3,910            | **16,335**               | × (fail to work)        |
> | 4,264            | **17,366**               | × (fail to work)        |
>
> Secondly, while GNN-based methods have been utilized in ride-sharing scenarios, they primarily extract local features (unless a dense graph is constructed, which would make it similar to a BERT-based encoder). This approach is not suitable for integration with our centralized framework. Furthermore, the construction of the graph in graph-based methods is crucial and directly influences model performance. However, this step requires careful design and can add computation overhead.
>
> > W3:  On Page 5, Line 269, this guarantees a unique solution. No proof or explanation on this point.
>
> Thank you for pointing it out. We acknowledge that this design cannot lead to a unique solution unless the hidden dimension is 1. Our design only avoid one particular redundancy scenario as mentioned in the paper, i.e. avoid the $f$ becomes $\alpha f$ while $g$ becomes $\frac{g}{\alpha}$. And the ablation study in Figure 3 also illustrates the efficacy of our our design, where without our normalization, the method fail to converge. We have revised the description in the manuscript. (Line 285-287 in Page 6)
>
> > W4: On Page 6, Line 323-324, the authors assume that unsatisfactory independent MARL can provide a good starting point. This point is not proved or analyzed to some extent.
>
> Thank you for your insightful question. Please refer to our reply to your comment in `W1` and the experimental results reported in Figure 3. We observe that without the MARL pre-trained starting point, it is difficult for the SARL method to train efficiently and find a good solution.

---

> ### Author Response · Authors · 2025-11-17
> **Response to Reviewer TZpd (Part 2)**
>
> > W5: On Page 6, Line 299-300, the authors claim that IDDQN is chosen for the stage-1 pre-training. However, in Table 1, the authors present the proposed Triple-BERT uses TD3 as a difference to comparison methods. It is a bit confusing since the whole framework is a two-stage approach and it is inappropriate to only highlight the stage-2 as a contribution.
>
> Thank you for pointing this out. We have revised Table 1 to clarify that our Triple-BERT is a two-stage method, which includes both the IDDQN-based MARL pre-training and the TD3-based SARL fine-tuning.
>
> ---
>
> Finally, we express our heartfelt gratitude for your insightful questions and suggestions. They have been invaluable to our work. We hope our efforts will adequately address your concerns.

---

### Official Review · Reviewer_TpDZ · 2025-10-31

**Soundness:** 3
**Presentation:** 3
**Contribution:** 2
**Rating:** 6
**Confidence:** 5

**Summary:**

This paper addresses the real-time order dispatching problem in large-scale ride-hailing platforms, where numerous drivers and passengers must be matched dynamically under uncertainty. The authors propose Triple-BERT, a centralized single-agent reinforcement learning approach designed to overcome the scalability and coordination limitations of multi-agent RL (MARL) frameworks. The method extends the TD3 algorithm with (i) an action decomposition strategy that factorizes joint action probabilities and (ii) a BERT-based attention architecture to handle large observation spaces efficiently via parameter reuse and contextual embedding of driver–order relationships.
Experiments on a real-world Manhattan ride-hailing dataset show improvements of roughly 12% in total performance metrics, with additional gains in served orders (+4.3%) and pickup times (–22%). Code and models are made publicly available.

**Strengths:**

1. Well-written and clearly structured: The paper is technically sound, logically organized, and easy to follow. The motivation for replacing MARL with a single-agent formulation is well articulated.

2. Practical relevance: Large-scale order dispatching remains a key benchmark problem in reinforcement learning for logistics and mobility; the paper contributes to this ongoing line of research.

3. Architectural innovation: The integration of a BERT-based network to manage relational structure and parameter reuse is elegant and practically valuable.

**Weaknesses:**

1. Limited discussion of related MARL coordination strategies: The paper does not engage with recent advances in local/global reward allocation or communication mechanisms that also address coordination challenges in MARL. Without such comparisons, it remains unclear whether the proposed single-agent framework is superior conceptually or primarily an engineering simplification.

2. Scope of novelty: While the combination of TD3 and BERT-based embedding is technically solid, the conceptual contribution lies mainly in neural architecture design rather than introducing a fundamentally new RL principle. The innovation should be framed more as an architectural contribution.

3. Statistical robustness: Results are averaged over only three random seeds, which limits statistical reliability. Given the inherent stochasticity of dispatching and RL training, more repetitions would strengthen confidence in the reported improvements.

4. Clarification on matching procedure: The algorithm ultimately uses a bipartite matching layer for stabilization, suggesting that global coordination is still enforced externally. The authors should clarify to what extent Triple-BERT learns full end-to-end dispatching policies versus providing improved matching scores.

**Questions:**

1. How does Triple-BERT compare to MARL approaches that use reward decomposition or credit assignment to improve coordination?

2. Could the authors elaborate on the role of the final bipartite matching step — is it part of the learning policy or a post-processing stabilization?

3. How sensitive are the results to hyperparameters and random seeds? Would results remain consistent under higher variance conditions?

4. Does the model generalize to other cities or datasets beyond Manhattan?

5. What are the main computational trade-offs of the BERT-based encoder compared to simpler attention or graph-based alternatives?

---

> ### Author Response · Authors · 2025-11-17
> **Response to Reviewer TpDZ (Part 1)**
>
> We are grateful for your thoughtful feedback. Your assessment is valuable for us to polish our paper. We are glad to reply to your questions.
>
> > W1: Limited discussion of related MARL coordination strategies: The paper does not engage with recent advances in local/global reward allocation or communication mechanisms that also address coordination challenges in MARL. Without such comparisons, it remains unclear whether the proposed single-agent framework is superior conceptually or primarily an engineering simplification.
>
> Thank you for your insightful comment. According to your suggestion, we have revised our Related Work section (Appendix F) to discuss the latest developments in order dispatch methods for ride-sharing tasks and cooperative MARL methods in detail. Specifically, for the cooperative MARL methods you mentioned, we examine various types, including independent learning, centralized critics, value decomposition, consensus, and communication mechanisms. Additionally, we illustrate the reward types (local/global) used in our comparative experiments in Table 1 for clarity.
>
> For centralized critics and value decomposition methods, they require a centralized critic network like MASAC or a global mixture network like QMIX, which face the curse of dimensionality (CoD) challenge in ride-sharing scenarios where the number of drivers can exceed hundreds or thousands. Consensus-based methods often rely on multiple rounds of communication, leading to practical challenges in the ride-sharing task, which has high demands for real-time performance. Communication-based methods also represent an open challenge, as determining  what content to communicate and to whom poses risks associated with credit assignment. Additionally, adapting these MARL methods to ride-sharing tasks is non-trivial, since an order cannot be assigned to multiple drivers, resulting in exclusivity in action among agents—an issue not typically faced by standard MARL methods.
>
> In contrast, our SARL method inherently resolves these challenges by capturing global information directly, facilitating better cooperation among drivers. Overall, we believe our method offers a novel conceptual paradigm for ride-sharing tasks, laying a solid foundation for future SARL-based framework development.
>
> > W2: Scope of novelty: While the combination of TD3 and BERT-based embedding is technically solid, the conceptual contribution lies mainly in neural architecture design rather than introducing a fundamentally new RL principle. The innovation should be framed more as an architectural contribution.
>
> Thank you for your insightful comment. We agree with the reviewer that the main innovation does not stem from a new RL pipeline. In fact, the primary contribution of this paper is that we propose the first centralized ride-sharing framework based on SARL, including: (i) a BERT-based network to efficiently capture global information for centralized decision; (ii) an action decomposition mechanism to address the large action space challenge in ride-sharing task; and (iii) a modification to the actor training of TD3 method to adapt our action decomposition mechanism. Overall, the contribution of this paper falls into the efficient centralized ride-sharing framework design, not just the neural network structure, even it's a part of our contribution.
>
> We hope this clarification address your concern sufficiently. Additionally, to avoid any potential mislead, we have revised our first point of contribution at the end of introduction, to illustrate our method is not a new RL pipeline but a novel ride-sharing framework.  (Line 18-22 in Page 1, Line 78-82 in Page 2)
>
> > W3: Statistical robustness: Results are averaged over only three random seeds, which limits statistical reliability. Given the inherent stochasticity of dispatching and RL training, more repetitions would strengthen confidence in the reported improvements.
>
> Thank you for your insightful suggestions. We have added two more repetitive experiments, and the results are illustrated in Figure 3, where our method continues to demonstrate SOTA performance.

---

> > ### Comment · Reviewer_TpDZ · 2025-11-19
> > **response to reviewer answers**
> >
> > I want to thank the authors for answering all of my questions. I think that the numerical evaluation is still limited, a standard design that is statistically more relevant would evaluate the learned policy on at least 20-50 trials. This does however not require retraining 20-50 times, quite the opposite is the case. What one would typically like to see is that the learned policy works well during inference over different demand realizations, i.e., once trained, the policy should be evaluated on demand data from different days that was not contained in the training data.
> > Without such proper experiments it is hard to evaluate the performance and robustness of the proposed algorithm.

---

> > > ### Author Response · Authors · 2025-11-20
> > >
> > > Thank you very much for your further comments and direction. We have evaluated the methods in additional trials, specifically utilizing all-day data from July 18, 2024, from High Volume For-Hire Vehicle (FHV) trip data, which has a different order distribution compared to the yellow-taxi data used for training. The 24-hour data leads to 48 episodes, each lasting 30 minutes, where the order volume varies from 734 to 5,989 in each episode. The experiment results are presented in the table below:
> > >
> > > | Method             | Reward               | Service Rate  | Delivery Time  | Detour Time   | Pickup Time   | Confirmation Time |
> > > | :----------------- | :------------------- | :------------ | :------------- | :------------ | :------------ | :---------------- |
> > > | DeepPool           | 11258.56±2578.68     | 0.76±0.18     | 13.93±0.93     | 2.54±1.48     | 10.19±0.78    | 0.09±0.07         |
> > > | BMG-Q              | 11899.39±2804.65     | 0.78±0.17     | 13.27±1.00     | 2.44±1.54     | 9.39±0.43     | 0.15±0.11         |
> > > | HIVES              | 11183.12±2458.29     | 0.78±0.18     | 13.72±1.48     | 2.87±1.90     | 9.77±0.68     | **0.05±0.03**     |
> > > | Enders et al.      | 10512.65±2744.34     | 0.78±0.17     | 14.12±0.48     | 3.06±0.43     | 9.92±0.55     | 1.37±0.99         |
> > > | CEVD               | 12556.74±3303.93     | 0.80±0.13     | **12.33±0.72** | **2.25±1.33** | 8.02±1.27     | 0.09±0.08         |
> > > | Triple-BERT (ours) | **14329.74±4627.26** | **0.88±0.11** | 13.07±0.61     | 2.78±0.92     | **7.02±0.88** | 0.34±0.32         |
> > >
> > > The experimental results and findings remain consistent with the manuscript; our Triple-BERT model achieves the highest reward by optimizing pickup time to improve the service rate, even though this may lead to higher delivery and detour times due to increased order bundling. Additionally, we noted that our method exhibits a higher standard deviation in rewards compared to others. This can be explained by the findings illustrated in Figure 6 of the manuscript. In low-order volume scenarios, the performance among different methods does not substantially differ, as all orders can be effectively served. However, in high-order volume scenarios, our Triple-BERT shows a significant advantage, leading to higher rewards. As a result, this can contribute to a greater standard deviation when aggregating all scenarios.
> > >
> > > We thank the reviewer again for helping us improve our work. We will add these results to the manuscript at the end of the discussion so that other reviewers can find our modifications according to the previous line numbers. If you have any further concerns or suggestions, please feel free to let us know.

---

> ### Author Response · Authors · 2025-11-17
> **Response to Reviewer TpDZ (Part 2)**
>
> > W4: Clarification on matching procedure: The algorithm ultimately uses a bipartite matching layer for stabilization, suggesting that global coordination is still enforced externally. The authors should clarify to what extent Triple-BERT learns full end-to-end dispatching policies versus providing improved matching scores.
>
> Thank you for your insightful comment. We would like to clarify that Triple-BERT is indeed an end-to-end approach that learns full dispatching policies for the ride-sharing platform; the bipartite matching layer serves as an intermediate step within the framework to efficiently determine the optimal action, rather than enforcing global coordination externally.
>
> To elaborate, in a standard Actor-Critic (AC)-based reinforcement learning framework, the optimal action during exploitation is selected by maximizing the learned action probability, which aligns with the action of highest expected utility in the logit model. In our case, this process can be formulated as (see Eq. 6 and Appendix B.2 in the manuscript):
>
> $$
> \arg \max_{A_t \in \psi(S_t)} \pi_{\Theta}^T(A_t|S_t) =  \arg \max_{A_t \in \psi(S_t)}  \text{z}\left(\prod_{i,j \in \text{h}(A_t)} \mathscr{P}_{i,j,t}\right)
> $$
>
> $$
> = \arg \max_{A_t \in \psi(S_t)} \sum_{i,j \in \text{h}(A_t)} \log \mathscr{P}_{i,j,t}
> $$
>
> Bipartite matching is used here as a computationally efficient method to solve this maximization problem, allowing us to find the best feasible and optimal action $A_t$ under the current state and neural network outputs. Thus, rather than externally imposing global coordination, bipartite matching facilitates efficient action selection within the learned policy.
>
> During exploration, we introduce random noise into $\mathscr{P}$ and use the same bipartite matching method to sample actions in a manner consistent with the learned distribution $\pi_{\Theta}^T(\cdot|S_t)$. While this is not a perfect equivalence, the injected noise ensures that actions with higher predicted probabilities are sampled more frequently, as confirmed by our robustness checks (see Appendix C), where Triple-BERT achieves state-of-the-art performance even under challenging noise settings such as the BSC noise.
>
> In summary, the bipartite matching process is an integral intermediate step in the Triple-BERT framework, enabling efficient computation of the optimal or sampled action at each step. This allows Triple-BERT to function as a true end-to-end dispatching policy, rather than merely providing matching scores.
>
> > Q1: How does Triple-BERT compare to MARL approaches that use reward decomposition or credit assignment to improve coordination?
>
> Thank you for your question. We acknowledge that some CTDE methods, such as QMIX and COMA, have addressed credit assignment to some extent. However, most of these methods require a centralized critic network, which struggles with the Curse of Dimensionality (CoD) in ride-sharing scenarios, as demonstrated by models like HIVES, CEVD, and the work by Enders et al. in our comparative experiments. While these methods can achieve better cooperation compared to purely local reward-based independent MARL methods like DeepPool and BMG-Q from a theoretical standpoint, our experiments show they actually perform at a similar level. This is largely due to the increased difficulty of training caused by the CoD.
>
> In contrast to MARL methods, SARL inherently avoids issues such as low cooperation, credit assignment, and lazy agents, given that there is only one agent involved, which provides greater theoretical convergence. Practically, the ride-sharing problem is a centralized decision-making task where the platform has access to all information regarding drivers and orders. Previous methods introduced MARL primarily to tackle the CoD. Our approach, however, aims to resolve these issues through the proposed action decomposition mechanism in conjunction with a BERT-based network structure under a SARL framework, thereby avoiding the complications associated with MARL altogether.
>
> > Q2: Could the authors elaborate on the role of the final bipartite matching step — is it part of the learning policy or a post-processing stabilization?
>
> Thank you for your question. The final bipartite matching step is indeed a part of the learning policy. For more details, please refer to our response to your comment in `W4`.

---

> ### Author Response · Authors · 2025-11-17
> **Response to Reviewer TpDZ (Part 3)**
>
> > Q3: How sensitive are the results to hyper-parameters and random seeds? Would results remain consistent under higher variance conditions?
>
> Thank you for your insightful questions. First, in our experiments, we did not set a specific random seed, and our results can be easily reproduced with the code we have provided.
>
> Second, regarding hyper-parameters, these relate to the network structure (such as network layers, hidden dimensions, dropout rates, etc.) and RL algorithms (including learning rates, batch sizes, and training frequencies of actor and critic). Exploring all these during the discussion period is quite challenging. Since our network is primarily based on BERT and the RL algorithm relies on TD3, the influence of hyper-parameters for these components has been extensively discussed in previous work. Thus, we do not revisit them in this paper.
>
> Given the unique features of our method, we introduce a noise-based exploration strategy for TD3, which is not common in the standard TD3 approach. As a result, we focus on the influence of different types of noise in Appendix C, comparing Gaussian noise, Uniform noise, and BSC noise. For our comparisons with other methods, we selected the worst-case noise type, i.e., BSC noise. Even under these conditions, our method still outperforms all others, demonstrating its promising performance without relying on specific hyper-parameter selections. If you have specific concerns about other hyper-parameters, please feel free to let us know, and we will conduct further experiments. Additionally, we report our detailed hyper-parameter settings in Table 2, with the code open-sourced to guarantee reproducibility.
>
> For high variance conditions, please refer to our expanded experiments in Appendix E.5. We compare our Triple-BERT against others in extreme high-concurrency scenarios, where the number of drivers reaches 1,500 and 2,000 while the order volume hits 6,775. The results show that Triple-BERT still outperforms others, achieving an accumulative reward improvement of 18.9% and 7.83% over the previous SOTA, CEVD.
>
> > Q4: Does the model generalize to other cities or datasets beyond Manhattan?
>
> Thank you for your insightful question. To further demonstrate the generalization of our method, we conducted experiments using High Volume For-Hire Vehicle (FHV) trip data from Queens, New York City (Line 1218-1237 in Page 23). Unlike the Manhattan data used in our previous experiment, Queens is a larger area but has a lower order density. This difference among orders can increase the complexity of determining which orders to bundle together for vehicle sharing.
>
> For our experiment, we selected data from 19:00 to 19:30 on July 17, 2024, which included 2,024 valid orders. To adapt to this new scenario, we set the driver count to 500. The detailed performance of various methods is presented in the following table, where our method continues to achieve SOTA performance, primarily by optimizing pickup times, which leads to a higher service rate and increased profits. Additionally, as expected, the pickup time in Queens is significantly higher than in Manhattan (around 5-9 minutes), owing to the more dispersed distribution of orders and drivers.
>
> | Method             | Reward      | Service Rate | Delivery Time | Detour Time | Pickup Time | Confirmation Time |
> | ------------------ | ----------- | ------------ | ------------- | ----------- | ----------- | ----------------- |
> | DeepPool           | 5222.85     | 0.64         | 11.24         | 1.84        | 12.30       | **0.21**          |
> | BMG-Q              | 5362.00     | 0.66         | 9.63          | 1.08        | 12.98       | 0.27              |
> | HIVES              | 3560.80     | 0.60         | **8.30**      | **0.36**    | 14.67       | 0.41              |
> | Enders et al.      | 4543.68     | 0.61         | 10.41         | 0.85        | 13.39       | 2.25              |
> | CEVD               | 4388.83     | 0.62         | 11.61         | 1.33        | 13.74       | 0.29              |
> | Triple-BERT (ours) | **5577.83** | **0.72**     | 9.07          | 0.90        | **11.32**   | 0.23              |

---

> ### Author Response · Authors · 2025-11-17
> **Response to Reviewer TpDZ (Part 4)**
>
> > Q5: What are the main computational trade-offs of the BERT-based encoder compared to simpler attention or graph-based alternatives?
>
> Thank you for your insightful question. For our BERT-based encoder, we have removed the positional embedding to ensure positional invariance. This allows our encoder to be viewed as consisting of self-attention layers alongside linear layers. The computational complexity for each layer (which includes one self-attention and one linear layer) can be expressed as $O(n^2d + nd^2)$, where $n$ is the input length and $d$ is the hidden dimension. In contrast, for graph-based methods like the GAT used in BMG-Q, the computational complexity is $O(|E|d + nd^2)$, where $|E|$ represents the number of edges. Consequently, the computational complexity of a BERT-based encoder is equivalent to that of GAT under a dense graph, where $|E|=n^2$.
>
> Even though graph-based encoders can reduce computational complexity in some contexts, we have chosen not to use them for the following reasons: (i) While graph-based methods have been utilized in ride-sharing scenarios, they primarily extract local features (unless a dense graph is constructed, which would make it similar to a BERT-based encoder). This is not suitable for integration with our centralized framework. Furthermore, most previous graph-based methods, such as BMG-Q, focus solely on the relationships among drivers, neglecting orders. (ii) The construction of the graph in graph-based methods is crucial and directly influences model performance. However, this step requires careful design and can add computation overhead. (iii) As shown in our experimental results in Appendix E.5, Table 6, the computation time for each step is less than 0.2 seconds, even under extremely high order volumes and driver counts. This computation time is significantly lower than the decision period, which is 1 minute. Therefore, we believe the computational cost of the BERT-based encoder is not a serious bottleneck for our method in practical applications.
>
> ---
>
> Finally, we express our heartfelt gratitude for your insightful questions and suggestions. They have been invaluable to our work. We hope our efforts will adequately address your concerns.

---

### Official Review · Reviewer_vF6k · 2025-11-01

**Soundness:** 3
**Presentation:** 3
**Contribution:** 3
**Rating:** 6
**Confidence:** 3

**Summary:**

This paper proposes Triple-BERT, a centralized single-agent reinforcement learning (SARL) framework for large-scale ride-sharing order dispatch. It addresses MARL limitations (e.g., poor cooperation, high dimensionality) via a BERT-based architecture with self-attention to model driver-order relationships globally, QK-attention to reduce computational complexity, and action decomposition to simplify joint decisions. A two-stage training strategy (decentralized pre-training + centralized fine-tuning) mitigates SARL sample scarcity. Experiments show Triple-BERT outperforms MARL baselines by ~15% in reward, serving more orders with lower latency (<0.2s), and better scalability to thousands of agents.

**Strengths:**

- BERT’s self-attention effectively captures global driver-order interactions, enhancing cooperation.
- QK-attention reduces complexity from multiplicative to additive, enabling real-time dispatch.
- Handles 1,500–2,000 drivers/orders without retraining, outperforming MARL in high-concurrency scenarios.
- Two-stage training resolves SARL data scarcity via MARL pre-training.

**Weaknesses:**

- Action decomposition relies on an approximate "independent action probability" assumption, which may not fully reflect inter-dependencies.

**Questions:**

What challenges might be encountered when deploying this framework in practical applications?

---

> ### Author Response · Authors · 2025-11-17
> **Response to Reviewer vF6k**
>
> We are grateful for your thoughtful feedback. Your assessment is valuable for us to polish our paper. We are glad to reply to your questions.
>
> > W1: Action decomposition relies on an approximate "independent action probability" assumption, which may not fully reflect inter-dependencies.
>
> Thank you for this insightful comment. First, we would like to clarify that our proposed BERT-based architecture is specifically designed to capture the global state of all drivers and couriers, thus effectively accounting for inter-dependencies through a centralized network. Unlike decentralized or independent-agent frameworks, our approach formulates a centralized RL model where the state representation explicitly includes the individual states of all couriers. The policy, therefore, is a mapping from the collective state to the joint action space. In this way, the interdependencies among agents are encoded within the neural network, which learns to represent and utilize these interactions. This centralized modeling of inter-agent dependencies is, in fact, one of the major strengths and innovations of our work.
>
> Second, we appreciate the opportunity to clarify the action decomposition approach within our RL algorithm. We have revised the manuscript to more precisely describe this point. Our method does not assume that each courier’s action probability is strictly independent. Rather, we introduce a structural assumption on the policy function to make the problem tractable, essentially searching over a structured subclass of policies. This is necessary because, in the ride-sharing context, the joint action space is extremely large—often exceeding $10^{30}$ possible combinations (as shown in Appendix A)—which makes it computationally infeasible to directly generate or optimize a full joint action probability vector as in standard discrete Actor-Critic (AC) methods.
>
> To address this, our action decomposition (see Eq. 5) expresses the policy as:
>
> $$
> \pi_{\Theta}^T(A_t|S_t) = \text{z} \left(\prod_{i,j \in \text{h}(A_t)} \mathscr{P}_{i,j,t} \right)
> $$
>
> The intuition behind this approach is that an action $A_t$ with a higher product of individual probabilities $\prod_{i,j \in \text{h}(A_t)} \mathscr{P}_{i,j,t}$ is more likely to be selected. This decomposition serves as a practical approximation, bridging the gap between the network’s output and the RL policy, and greatly simplifying both optimization and action sampling (see Section 3.3.2 (i), Actor part). While this approach does impose some restrictions on the representational capacity of the actor, we have found that the benefits of our centralized architecture—especially in terms of explicitly modeling inter-agent dependencies—far outweigh these limitations. This is further supported by the state-of-the-art performance achieved by our Triple-BERT model.
>
> We have revised the relevant sections in the manuscript (Lines 363-390 on Pages 7-8, and Lines 894-904 on Page 17) to clarify these points and better reflect your suggestions. Thank you again for your valuable feedback, which has helped us improve the clarity and rigor of our work.
>
> > Q1: What challenges might be encountered when deploying this framework in practical applications?
>
> Thank you for your insightful question. As a centralized SARL method, our approach demonstrates significant performance improvements. However, it requires full observation of all drivers and orders, which can introduce certain challenges. In particular, it may not be as robust as traditional MARL methods in handling single points of failure. For example, if information from a particular driver is missing at any time step, the system’s operation could be compromised. One potential remedy is to assign the unavailable driver's state as its previous state while explicitly marking it as "unavailable." Nonetheless, developing efficient strategies to address this limitation remains an open area of research. We have now acknowledged and discussed this issue at the end of the conclusion (Lines 512-519, Page 10).
>
> ---
>
> Finally, we express our heartfelt gratitude for your insightful questions and suggestions. They have been invaluable to our work. We hope our efforts will adequately address your concerns.

---

> > ### Comment · Reviewer_vF6k · 2025-11-27
> >
> > Thanks for the authors' response. My concerns have been properly addressed and I decide to raise my score to 8.

---

> > > ### Author Response · Authors · 2025-11-27
> > >
> > > Dear Reviewer vF6k,
> > >
> > > Thank you once again for your thorough review and constructive feedback regarding our submission. It is our honor to address your concerns, and we are highly encouraged by your decision to raise our rating.
> > >
> > > Best Regards,
> > > ICLR 2026 Submission 11772 Author

---

### Author Response · Authors · 2025-11-29
**Rebuttal Summary**

Dear Reviewers, Area Chairs, and Program Committee Members,

We would like to thank the ACs and PCs for handling our paper, and we extend our gratitude to the reviewers for their expertise, effort, and time in reviewing our submission. Your invaluable feedback has greatly helped us improve our work. As the discussion has concluded, we summarize the acknowledged strengths highlighted by the reviewers and our responses to their concerns as follows.

---

**Strengths**:

* **Clear Structure and Motivation with Practical Importance**: Our work focuses on the order dispatch task in ride-sharing platforms, which is significant for platform revenues, passenger travel efficiency, and reducing carbon emissions. We identify the low cooperation situation of most previous MARL methods and propose to address this issue using Single-Agent Reinforcement Learning (SARL), representing a fundamental shift. `TpDZ` `TZpd` `PcM6`

* **Novel Model Framework with Clear Writing**: To implement the SARL-based ride-sharing framework, we propose a novel BERT-based network that utilizes global information. The parameter reuse in BERT also addresses the Curse of Dimensionality (CoD) challenge. Additionally, we propose a novel action decomposition mechanism to manage the extremely large action space. `vF6k` `TpDZ` `PcM6`

* **Promising Performance**: Through experiments on a real-world dataset, our method achieved SOTA performance across multiple scenarios and efficiently scales to scenarios with more vehicles and orders without the need for retraining. `vF6k`

---

**Concerns and Our Response**:

* **Insufficient Experimentation**: We have further evaluated our method using an entire day’s data from July 18, 2024, from High Volume For-Hire Vehicle (FHV) trip data, resulting in 48 independent repetitive episodes. Additionally, we tested the method with data from another city, Queens in New York. All these experiments show that our proposed method maintains SOTA performance, demonstrating its high generalization and robustness. `TpDZ`

* **Limited Literature Review**: We have revised the related work section in the appendix, providing a detailed review of literature on cooperative MARL and ride-sharing methods while outlining the advantages of our approach. In summary, the ride-sharing task is fundamentally a centralized decision task, and our method, based on the SARL framework, avoids potential shortcomings associated with MARL. `vF6k` `PcM6`

* **Writing Improvement**: We have revised the writing regarding our contribution claims, methodology descriptions (including QK-attention and the action decomposition mechanism), and the limitations and future directions section.`vF6k` `TZpd` `PcM6`

---

Finally, we express our gratitude once again for your contributions to this conference.

Sincerely,

ICLR 2026 Submission 11772 Authors

---

### Meta-Review · Area_Chair_MCCZ · 2025-12-10

**Summary:**

Well-motivated and clearly written paper proposing a centralized SARL framework (Triple-BERT) with a BERT-based encoder and action decomposition for large-scale ride-sharing order dispatch, supported by strong empirical results on real-world data (including added cross-day and cross-city evaluations). Reviewers appreciate the practical impact, architectural design, and thorough rebuttal, though they note that the novelty is primarily architectural, the theoretical analysis of action decomposition is limited, and the “no MARL” messaging is somewhat overstated given the MARL pretraining stage.

The authors must better calibrate novelty claims, clarify the MARL–SARL positioning, and surface key related-work and design-justification discussions into the main text.

**Reviewer Concerns:**

Most reviewer concerns were effectively addressed in the rebuttal and discussion.
In particular, concerns about (i) insufficient experiments and statistical robustness, (ii) limited related-work coverage (especially MARL coordination and ride-pooling literature), (iii) clarity of the action-decomposition mechanism, and (iv) the role of bipartite matching in end-to-end learning were all satisfactorily resolved with additional analyses, new experiments (full-day + Queens), expanded explanations, and revisions.
The remaining outstanding issues are (a) the broader conceptual-novelty question (the contribution is still primarily architectural), (b) the title/narrative tension around “not needing MARL” despite relying on MARL pretraining, and (c) the lack of deeper theoretical guarantees for the action decomposition beyond heuristic justification.

**Reviewer Scores:**

The scores are already good. One reviewer is mentioning increasing from 6 to 8.

---

### Decision · Program_Chairs · 2026-01-26

Accept (Oral)